# NDUFS4 regulates cristae remodeling in diabetic kidney disease

Koki Mise [1,2], Jianyin Long[1], Daniel L. Galvan [3], Zengchun Ye[4], Guizhen Fan[5], Rajesh Sharma[5], Irina I. Serysheva [5], Travis I. Moore [6], Collene R. Jeter[7], M. Anna Zal[7], Motoo Araki[8], Jun Wada [2], Paul T. Schumacker[9], Benny H. Chang[1] & Farhad R. Danesh [1,10] ✉

The mitochondrial electron transport chain (ETC) is a highly adaptive process to meet metabolic demands of the cell, and its dysregulation has been associated with diverse clinical pathologies. However, the role and nature of impaired ETC in kidney diseases remains poorly understood. Here, we generate diabetic mice with podocyte-specific overexpression of Ndufs4, an accessory subunit of mitochondrial complex I, as a model investigate the role of ETC integrity in diabetic kidney disease (DKD). We find that conditional male mice with genetic overexpression of Ndufs4 exhibit significant improvements in cristae morphology, mitochondrial dynamics, and albuminuria. By coupling proximity labeling with super-resolution imaging, we also identify the role of cristae shaping protein STOML2 in linking NDUFS4 with improved cristae morphology. Together, we provide the evidence on the central role of NDUFS4 as a regulator of cristae remodeling and mitochondrial function in kidney podocytes. We propose that targeting NDUFS4 represents a promising approach to slow the progression of DKD.

Diabetic kidney disease (DKD), a serious microvascular complication of diabetes, is the leading cause of end-stage kidney disease worldwide[1]. Recent studies have shown promising results with the use of sodium-glucose cotransporter-2 (SGLT2) inhibitors and glucagon-like peptide-1 (GLP-1) receptor agonists in reducing the risk of cardiovascular disease and progression of DKD[2,3]. However, despite recent advances, the risk of progression to renal replacement therapy and kidney transplantation in individuals with DKD remains high, underscoring the need for continued research into the underlying molecular mechanisms involved in the pathobiology of DKD and the development of novel therapies to address this unmet need.

Among multiple molecular mechanisms that have been implicated in the pathogenesis of DKD, the role of mitochondrial dysfunction in DKD has become increasingly recognized[4–7]. Indeed, several recent studies have identified a spectrum of biochemical and structural abnormalities of mitochondria in the diabetic environment, including a significant increase in generation of reactive oxygen species (ROS), changes in mitochondrial biogenesis, bioenergetics and mitochondrial dynamics[8–10]. Mitochondria are dynamic organelles that

[1]Section of Nephrology, The University of Texas MD Anderson Cancer Center, Houston, TX, USA. [2]Department of Nephrology, Rheumatology, Endocrinology & Metabolism, Okayama University Graduate School of Medicine, Dentistry & Pharmaceutical Sciences, Okayama, Japan. [3]Department of Hematopathology, The University of Texas MD Anderson Cancer Center, Houston, TX, USA. [4]Division of Nephrology, The Third Affiliated Hospital of Sun Yat-Sen University, Guangzhou, China. [5]Department of Biochemistry and Molecular Biology, The University of Texas Health Science Center at Houston, Houston, TX, USA. [6]Department of Integrative Biology and Pharmacology, The University of Texas Health Science Center at Houston, Houston, TX, USA. [7]Department of Epigenetics and Molecular Carcinogenesis, The University of Texas MD Anderson Cancer Center, Houston, TX, USA. [8]Department of Urology, Okayama University Graduate School of Medicine, Dentistry & Pharmaceutical Sciences, Okayama, Japan. [9]Department of Pediatrics, Feinberg School of Medicine, Northwestern University, Chicago, IL, USA. [10]Department of Biochemistry and Molecular Pharmacology, Baylor College of Medicine, Houston, TX, USA. ✉e-mail: fdanesh@mdanderson.org

are known to change their shape, number, distribution and metabolism in response to changes in their cellular environment[11,12]. Consistent with a critical role of mitochondrial dynamics in the progression of DKD, we have previously shown that kidney podocytes undergo a shift towards fragmented mitochondria and enhanced mitochondrial fission through a signaling cascade that involves dynamin related protein-1 (Drp1) phosphorylation at serine 600 residue and recruitment of Drp1 to the mitochondria that ultimately leads to increased production of mitochondrial ROS (mROS) and the progression of DKD[13,14]. However, despite considerable progress in understanding the role of mitochondrial morphology and function in kidney cells[15,16], the exact role and nature of mitochondrial dysfunction in the diabetic environment remains largely unknown.

The "unifying theory of microvascular complications of diabetes", a concept conceived by Brownlee and colleagues, proposed that mitochondrial dysfunction in microvascular complications of diabetes could be defined as an excess production of mROS through impaired electron transport chain (ETC) function[17,18]. However, the conclusions drawn from this study were widely challenged because of the lack of in vivo evidence to link mitochondrial respiration defects with mROS generation and the pathogenesis of DKD.

Maintenance of mitochondrial respiration requires the proper assembly and function of the ETC, composed of four protein complexes (complex I–IV; CI–IV) embedded within the inner membrane of the mitochondrion to allow for the transfer of electrons from the substrates to oxygen and generation of an electrochemical gradient[12]. Mitochondrial complex I (CI, NADH: ubiquinone oxidoreductase) constitutes the first and the largest component of ETC (-1 MDa) that couples the transfer of two electrons from NADH to ubiquinone with the translocation of four protons across the inner mitochondrial membrane[19,20]. In mammals, CI is a highly organized L-shaped holoenzyme that is comprised of 14 conserved catalytic "core subunits" and 31 "accessory or supernumerary subunits"[21]. Recent evidence suggests that CI can exist independently as a single entity or as part of stable high-molecular weight respiratory supercomplexes (RSCs) composed of CI, complex III (CIII), and complex IV (CIV) in varying stoichiometric ratios[22]. The functional relevance of RSCs remains elusive but proposed roles of RSCs include enhancing the structural stability of individual complexes and increasing respiration rate and decreasing electron leak and generation of mROS[23,24].

ETC dysfunction has emerged as an important cause of organ failure in several human pathologies including heart failure, diabetes, neurodegenerative diseases and kidney failure[25–30]. Despite this recognition, the precise implications of impaired ETC function and the impact of targeting CI in kidney pathologies remain largely unknown. In the current study, we investigated whether and how ETC remodeling in kidney podocytes contributes to the progression of DKD. We initially observed that the expression of several subunits of mitochondrial CI were significantly reduced in the kidney podocytes of diabetic models of DKD. To determine whether these changes are pathogenic or adaptive in response to the diabetic milieu, we engineered a podocyte-specific Ndufs4 (*NADH: ubiquinone oxidoreductase iron-sulfur protein 4*) transgenic mouse model. NDUFS4, an accessory 18 kDa subunit of the CI, is the most extensively studied CI subunit as a model for mitochondrial diseases. We find that diabetic podocyte-specific Ndufs4 transgenic mice exhibit improved CI activity associated with marked improvements in cristae morphology and mitochondrial dynamics. Our findings provide evidence that Ndufs4 ties the proper activity and function of CI to high glucose metabolic cues in the cell and the progression of DKD.

## Results

### ETC remodeling of podocytes in DKD

ETC is known to exhibit significant remodeling in response to the metabolic demands of the cell[19]. However, it remains unclear how ETC adapts to the diabetic milieu in podocytes. To provide a deeper understanding of the possible remodeling of ETC complexes in the diabetic environment, we performed a comparative mitochondrial proteome profiling focusing quantitatively on the protein abundance of ETC complexes in primary podocytes isolated from diabetic C57BL/6-*Ins2^{Akita}*/J (*Ins2^{Akita/+}*) mice, an established model of type 1 diabetes, and their nondiabetic littermates (Fig. 1a, Supplementary Fig. 1a). We identified 61 out of the 73 known subunits of mouse ETCs corresponding to a recovery rate of 84% (Fig. 1b). Notably, we found that the abundance of several subunits of CI was reduced in the podocytes of diabetic mice (Fig. 1b, c, Supplementary Fig. 1b). Reduction of CI subunits were also observed in a comparative mitochondrial proteomic profiling in primary podocytes isolated from type 2 diabetic *Lepr^{db/db}* compared with their control *Lepr^{db/+}* mice (Fig. 1e, Supplementary Fig. 1c). Consistent with these findings, we also observed reduced rotenone-sensitive CI enzymatic activity in enriched mitochondrial samples from podocytes in both type 1 (*Ins2^{Akita/+}*) and type 2 (*Lepr^{db/db}*) diabetic mice (Fig. 1d, Supplementary Fig. 1d). Among the most prominently reduced CI subunits validated in a series of quantitative RT-PCR experiments (Fig. 1e, Supplementary Fig. 1e, f), we focused on Ndufs4, an accessory 18 kDa subunit of the CI, for further analysis since its mRNA and protein levels were consistently reduced not only in podocytes of type 1 (*Ins2^{Akita/+}*) and type 2 (*Lepr^{db/db}*) diabetic mouse models (Fig. 1e, g, Supplementary Fig. 1e, f, h) but also in the glomeruli of subjects with DKD (Fig. 1e, Supplementary Fig. 1g), whereas several other potential CI candidates, including NDUFA2, NDUFB3, NDUFB4, NDUFB5, NDUFB8, NDUFB11, and NDUFV3 were not consistently downregulated in the diabetic mice or in the glomeruli of patients with DKD (Supplementary Fig. 1e–g). Interestingly, NDUFS4 protein expression in the kidney tubular cells remains unchanged in diabetic mice (Supplementary Fig. 1h). Using Nephroseq database, we found a positive correlation between *NDUFS4* mRNA in the glomeruli and estimated glomerular filtration rate (eGFR) in subjects with DKD (Fig. 1h). We validated these findings by immunohistochemistry (IHC) on paraffin sections in a cohort of 34 subjects with biopsy-proven DKD (Supplementary Fig. 1i, Supplementary Table 1). We found that the glomerular NDUFS4 expression correlated positively with eGFR ($r^2 = 0.289$, $P = 0.001$; Fig. 1i) and negatively with the urinary albumin excretion rate (UACR) ($r^2 = 0.219$, $P = 0.005$; Fig. 1j). Importantly, NDUFS4 staining was significantly decreased in the kidney glomeruli of diabetic subjects with normoalbuminuria as compared to that of healthy donors (5.8 ± 0.7 *vs.* 14.0 ± 1.9 pixel/μm², $P < 0.001$; Fig. 1k), suggesting that NDUFS4 downregulation may have occurred prior to the clinical manifestation of DKD. We also evaluated NDUFS4 staining in the glomeruli from diabetic subjects with a wide spectrum of DKD histology[31], and found that NDUFS4 staining in glomeruli was progressively reduced with worsening of DKD histology (test for trend $P < 0.01$) (Fig. 1l). Taken together, the consistently reduced expression of Ndufs4 in both murine models of diabetes and in human DKD subjects suggest that Ndufs4 may play an important role in progression of DKD.

### Podocyte-specific overexpression of Ndufs4 mitigates the progression of DKD

To assess the potential role of reduced NDUFS4 in the development of DKD, we engineered a podocyte-specific Ndufs4 transgenic mouse model (*Ndufs4^{PodTg}*) (Fig. 2a). Hemizygous *Ndufs4^{PodTg}* mice were indistinguishable from wild-type (WT) mice (Fig. 2a). Compared to podocytes isolated from age-matched littermate WT mice, there was -10-fold higher mRNA expression and -2-fold increase in protein expression of Ndufs4 in primary podocytes of *Ndufs4^{PodTg}* mice (Fig. 2b, c). Notably, CI activity as measured enzymatically using mitochondria isolated from primary podocytes was ~20% higher in *Ndufs4^{PodTg}* compared to the WT mice (Fig. 2d). Similarly, as assessed by NADH oxidase staining[32], we observed a - 30% increase in CI activity in the glomeruli

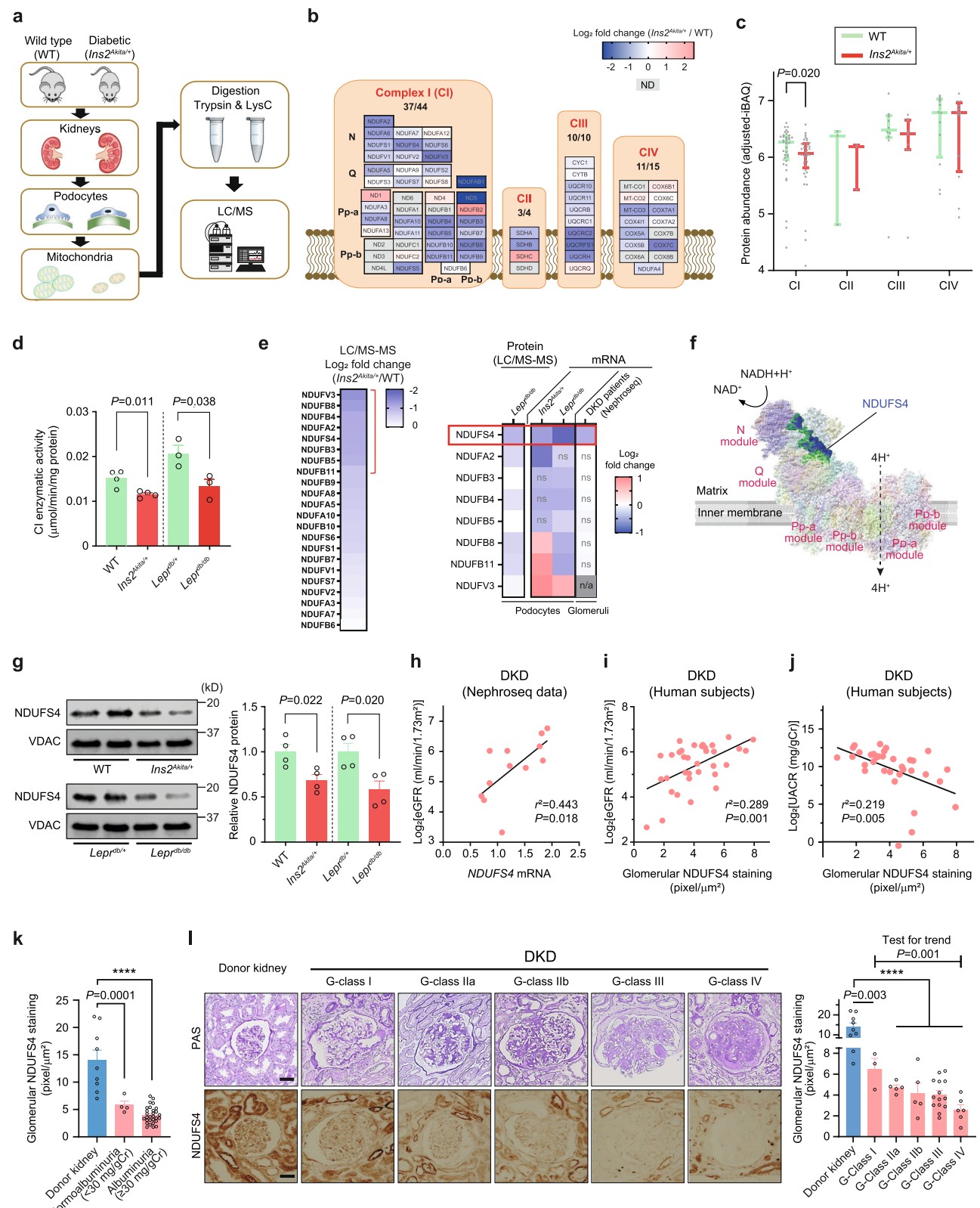

of the kidney tissue sections in the *Ndufs4^{PodTg}* mice compared to the WT (Supplementary Fig. 2a). We next crossed *Ndufs4^{PodTg}* mice with diabetic *Ins2^{Akita/+}* mice to generate diabetic *Ins2^{Akita/+};Ndufs4^{PodTg}* (herein diabetic *Ndufs4^{PodTg}*) mice. Diabetic *Ndufs4^{PodTg}* mice do not differ in body weight, blood glucose, and HbA1c when compared to diabetic *Ins2^{Akita/+}* mice (Fig. 2e, f, Supplementary Fig. 2b–d). However, they exhibit significant protection against biochemical and histological

features of DKD, including albuminuria (Fig. 2g, h and Supplementary Fig. 2e, f), kidney and glomerular hypertrophy (Fig. 2i, j), mesangial matrix expansion (Fig. 2k), glomerular basement membrane (GBM) thickening (Fig. 2l), podocyte foot process effacement and podocytes loss (Fig. 2i, m). We also crossed *Ndufs4^{PodTg}* mice with obese *Lepr^{db/db}* mice, an established model of type 2 diabetes, and found similar phenotypes with a significant improvement in albuminuria in diabetic

**Fig. 1 | Reduced Ndufs4 expression in podocytes in DKD. a** Experimental workflow of quantification and comparison of mitoproteomes in primary mouse podocytes from diabetic $Ins2^{Akita/+}$ vs. WT mice (pooled samples from 8 mice/group). **b** Heatmap illustrating $\log_2$ fold change in mitochondrial ETC protein abundance in diabetic $Ins2^{Akita/+}$ vs. WT mice. **c** Aggregated abundance of ETC proteins in podocytes of WT vs. diabetic $Ins2^{Akita/+}$ mice ($n = 37$ (CI), $n = 3$ (CII), $n = 10$ (CIII), $n = 12$ (CIV) subunits/group). Data are presented as median ± inter quartile range (IQR). **d** Rotenone-sensitive CI enzymatic activity in mitochondrial enriched samples isolated from primary podocytes ($n = 3$ (WT and $Ins2^{Akita/+}$), $n = 4$ ($Lepr^{db/m}$ and $Lepr^{db/db}$) independent experiments), where each sample was derived from a pool of podocyte mitochondria isolated from 4–6 mice. **e** Mitoproteomes of the most downregulated CI subunits in the podocytes of the diabetic $Ins2^{Akita/+}$ mice (left panel) and a selection of reduced CI subunits (shown in Fig. 1b) validated by podocyte mitoproteomes from diabetic $Lepr^{db/db}$ mice (left lane on the right panel), qRT-PCR (2 center lanes on the right panel) of two murine diabetic models, and mRNA of CI subunits in diabetic patients with DKD obtained from Nephroseq v5 dataset (nephroseq.org; Ju CKD glom database). ns: not significant and n/a: not available. **f** Human CI structure highlighting the location of NDUFS4 (blue) in the N module of the matrix arm of CI. The structure visualization was rendered through UCSF ChimeraX software (www.rbvi.ucsf.edu/chimerax). **g** Left panel,

representative immunoblots of NDUFS4 using mitochondria-enriched samples from primary podocytes of WT, $Ins2^{Akita/+}$, $Lepr^{db/+}$, and $Lepr^{db/db}$ mice. VDAC was used as a loading control. Right panel, densitometric analysis of western blots normalized to control levels ($n = 4$, each $n$ represents a pool of mitochondria from 4 different mice). **h** Pearson's correlation analysis of human $NDUFS4$ mRNA expression. Data obtained from Nephroseq v5 ($n = 12$). Median-centered $\log_2$ values of eGFR are used for the analysis. **i, j** Pearson's correlation of glomerular NDUFS4 immunostaining with eGFR (**i**) and urinary UACR (**j**) from diabetic subjects with DKD ($n = 34$). **k** Representative NDUFS4 immunostaining in glomeruli obtained from biopsies from healthy donors ($n = 9$) and DKD patients with ($n = 29$) or without albuminuria ($n = 4$). **l** Glomerular NDUFS4 immunostaining in biopsies from healthy donors and DKD individuals with different stages of glomerular involvement: Donors ($n = 9$), Glomerular (G)-class I DKD ($n = 3$), G-class IIa DKD ($n = 5$), G-class IIb DKD ($n = 5$), G-class III DKD ($n = 14$), G-class IV DKD ($n = 6$), Scale bars = 50 μm. Data are presented as mean ± SEM (**d, g, k, l**). ****$P < 0.0001$ by paired two-tailed $t$-test followed by Holm-Sidak test (**c**), unpaired two-tailed $t$ test (**d,g**), two-tailed test for Pearson's correlation (**h–j**), one-way ANOVA with post-hoc Tukey–Kramer test (**k, l**), and test for linear trend analysis for different classifications of DKD (**l**). Source data are provided as a Source Data file.

---

$Lepr^{db/db};Ndufs4^{PodTg}$ independent of body weight gain and blood glucose levels (Supplementary Fig. 2g–k).

## Ndufs4 overexpression improves mitochondrial morphology

We reasoned that the underlying molecular mechanism of $Ndufs4^{PodTg}$-mediated improvement in DKD could be associated with improved mitochondrial respiration in podocytes. To test this, we measured mitochondrial respiration using a Seahorse Analyzer. Whereas the oxygen-consumption-rate (OCR) measurements were similar between primary podocytes from WT and $Ndufs4^{PodTg}$ mice, podocytes from diabetic $Ins2^{Akita/+}$ mice exhibited a significantly reduced basal, maximal, ATP-linked, and spare OCR values (Fig. 3a–e). In contrast, OCR values were markedly improved in podocytes from diabetic $Ndufs4^{PodTg}$ mice (Fig. 3a–e). We also determined the respiratory control ratio (RCR) and found that podocytes from diabetic $Ins2^{Akita/+}$ mice exhibited lowered RCR values as compared to WT and non-diabetic $Ndufs4^{PodTg}$ mice ($7.86 ± 1.02$ vs. $11.34 ± 0.59$ and $11.46 ± 0.52$, respectively) but these values were restored in podocytes from diabetic $Ndufs4^{PodTg}$ mice ($11.10 ± 0.87$) (Supplementary Fig. 3a). To further examine podocytes energy metabolism shifts under the diabetic condition with Ndufs4 overexpression, we generated energy maps based on the OCR and the EACR values under basal and maximal respirations (Supplementary Fig. 3b, c). Our analysis revealed a distinct metabolic profile in podocytes from diabetic $Ins2^{Akita/+}$ mice, characterized by a reduced metabolic activity compared to their WT counterparts. In contrast, Ndufs4 overexpression in podocytes from $Ins2^{Akita/+}$ mice prompted a shift in energy metabolism towards a more robust state. This shift closely resembled the energetic profile observed in WT cells, featuring increased mitochondrial oxidative phosphorylation and glycolysis (Supplementary Fig. 3b, c).

We next tested the susceptibility of these podocytes to rotenone, a CI-specific inhibitor. We found that although podocytes from diabetic $Ins2^{Akita/+}$ mice had a significantly lower OCR suppression curve and rotenone $IC_{50}$ values, primary podocytes from diabetic $Ndufs4^{PodTg}$ mice had much higher $IC_{50}$ and OCR suppression curve, almost similar to those in WT mice (Fig. 3f). Consistent with these findings, we also found improved CI activity assessed by NADH oxidase staining in the glomeruli of the kidney tissue sections from the diabetic $Ndufs4^{PodTg}$ mice compared to that in podocytes from $Ins2^{Akita/+}$ mice (Fig. 3g).

We have previously shown that kidney podocytes exposed to high glucose conditions exhibit a fragmented or punctate phenotype that contributes to the progression of DKD[13,14]. Therefore, we next explored the contributions of $Ndufs4^{PodTg}$ on mitochondrial dynamics. We confirmed that podocytes from type 1 $Ins2^{Akita/+}$ and type 2 $Lepr^{db/db}$ diabetic

mice exhibited enhanced mitochondrial fission with altered cristae morphology (Fig. 3h–n, Supplementary Fig. 3d–m). However, over-expression of NDUFS4 restored the tubular and interconnected mitochondrial morphology as well as cristae density as shown by improved mitochondrial aspect ratio, circularity, roundness, ferret measurements, and cristae number and junctions in podocytes (Fig. 3h–n, Supplementary Fig. 3d–m).

To assess the mROS, we used the mito-roGFP2, a mitochondrial matrix targeted ratiometric redox-sensitive green fluorescent protein as previously described by our groups and others[33,34]. We measured mito-roGFP ratios in live mitochondria using confocal microscopy, in live cells. We found that primary podocytes from diabetic mice show a significant increase in oxidized/reduced mito-roGFP ratios compared to those from WT mice (Fig. 3o), suggesting that mitochondrial thiol oxidation increases in diabetic podocytes. On the other hand, primary podocytes from diabetic mice overexpressing Ndufs4 show significantly reduced oxidized/reduced mito-roGFP ratios compared with podocytes from diabetic mice, indicating Ndufs4 overexpression prevents enhanced mROS in diabetes. Similar results were obtained using a fluorogenic dye MitoSOX (Supplementary Fig. 3n). In addition, primary podocytes from diabetic $Ndufs4^{PodTg}$ mice exhibited enhanced ATP production (Fig. 3p, q) and total ATP content (Supplementary Fig. 3o), compared to those from the diabetic $Ins2^{Akita/+}$ mice.

## Forced expression of Ndufs4 prevents cristae remodeling in podocytes

To further interrogate the causal link between Ndufs4 overexpression (OE) and mitochondrial reprogramming, we generated a $PiggyBac$ transposon vector stably expressing a doxycycline (DOX)-inducible Ndufs4 expression (NDUFS4 OE) cassette (Supplementary Fig. 4a, b). We first excluded a direct effect of DOX (200 nM) on mitochondrial function (Supplementary Fig. 4c–e). Similar to our previous results with primary podocytes, we found enhanced mitochondrial fission and a significant cristae remodeling characterized by reduced cristae density in differentiated podocytes treated with HG (25 mM for 48 hrs), whereas cristae integrity was maintained in DOX-induced NDUFS4 OE in HG stress conditions, mainly resembling cristae morphology in NG conditions (5.5 mM for 48 h) (Fig. 4a–d). We also validated prevention of HG-induced reduction in mitochondrial CI enzymatic activity (Supplementary Fig. 4f, g) and improvement of ATP content in NDUFS4 OE podocytes cultured under HG conditions (Supplementary Fig. 4h).

Since several publications have recently established the link between intact cristae morphology, the stability of respiratory super-complexes (RSC), and mitochondrial morphology[35–38], we argued that

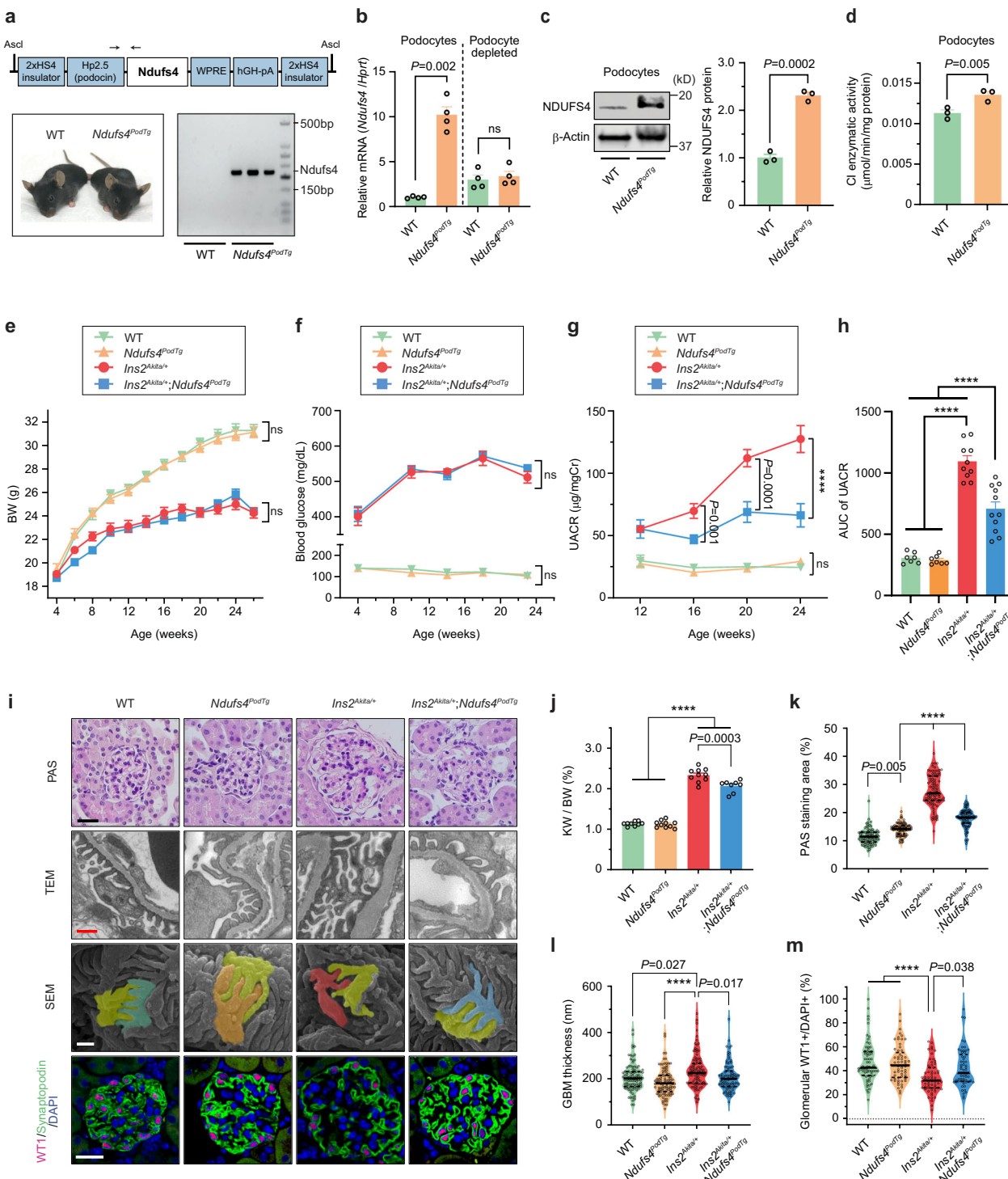

**Fig. 2 | Podocyte-specific Ndufs4 overexpression ameliorates DKD progression.**
**a** Schematic depiction of the construct used to engineer podocyte-specific Ndufs4-transgenic mice (*Ndufs4^PodTg^*) (top), representative images of WT and transgenic mice (lower left), and PCR genotyping (lower right). **b** qRT-PCR of *Ndufs4* mRNA in primary podocytes and podocytes-depleted samples isolated from 8-week-old WT and *Ndufs4^PodTg^* mice (*n* = 4). **c** A representative NDUFS4 immunoblot (left) and quantification of NDUFS4 protein expression (right) (*n* = 3). **d** Rotenone-sensitive CI enzymatic activity in mitochondrial-enriched samples isolated from primary podocytes of 8-week-old WT and *Ndufs4^PodTg^* mice (*n* = 3). Body weight (BW) (**e**, *n* = 12 mice/group), blood glucose (**f**, *n* = 8 (WT and *Ndufs4^PodTg^*), *n* = 10 (*Ins2^Akita/+^*), *n* = 11 (*Ins2^Akita/+^;Ndufs4^PodTg^*)), and UACR (**g**) in mice. **h** Area under the curve (AUC) of UACR shown in **g** (**g**, **h**, *n* = 7 (WT and *Ndufs4^PodTg^*), *n* = 10 (*Ins2^Akita/+^*), *n* = 11 (*Ins2^Akita/+^;Ndufs4^PodTg^*)). **i** Representative micrographs of Periodic acid-Schiff (PAS) stained

kidney sections, podocyte morphology in transmission electron microscopy (TEM), scanning electron microscopy (SEM), and immunostaining of Wilms' tumor 1 (WT1) (pink), Synaptopodin (green), and DAPI (blue). Scale bars = 50 μm in rows 1 and 4; 500 nm in rows 2 and 3. **j** Kidney weight (KW) per body weight (BW) of 26-week-old mice (*n* = 9 (WT), *n* = 10 (*Ndufs4^PodTg^* and *Ins2^Akita/+^*), *n* = 8 (*Ndufs4^PodTg^;Ins2^Akita/+^*)). **k** PAS positive area in glomeruli (*n* = 82–96 from 3 mice/group). **l** Glomerular basement membrane thickness (*n* = 100 areas of TEM images from 3 mice/group). **m** WT1 positive nuclei in glomeruli (*n* = 90–100 from 3 mice/group). Data are presented as mean ± SEM (**b–h**, **j**) or median ± IQR (**k–m**, bold line: median, and dot line: IQR). ns not significant; ****P < 0.0001; unpaired two-tailed Welch's *t* test (**b–d**), one-way ANOVA with post-hoc Tukey–Kramer test (**e–h**, **j**), or Kruskal–Wallis with post-hoc Dunn's test (**k–m**). Source data are provided as a Source Data file.

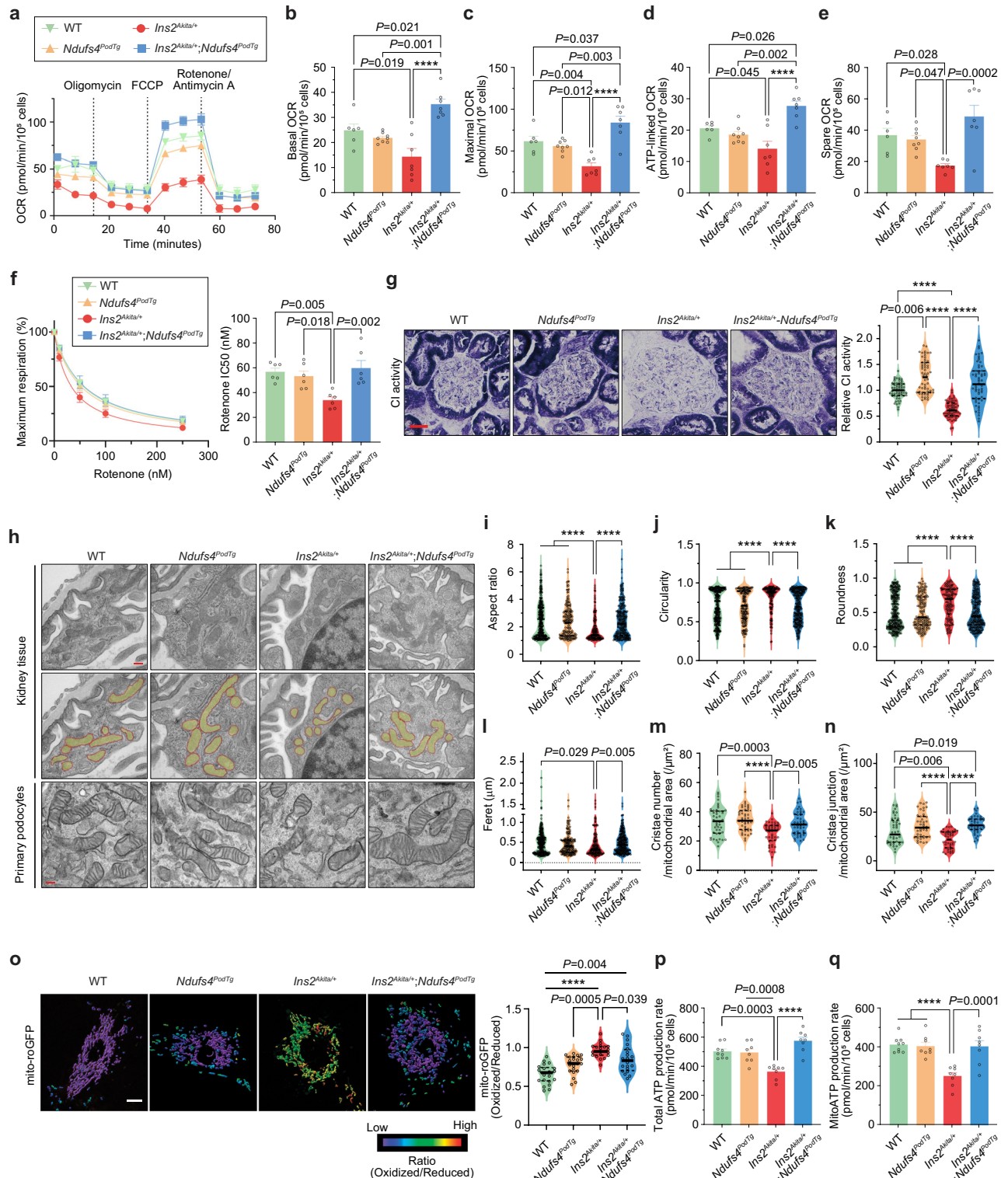

a possible relationship between NDUFS4 OE and improved cristae integrity could explain some of the beneficial effects of NDUFS4 OE on mitochondrial morphology. To this end, we first validated our observations using electron cryo-tomography (cryo-ET) of cristae structures and found substantial differences in cristae morphology consistent with a significant loss of cristae membranes in HG-exposed podocytes (Fig. 4e), whereas transfection of podocytes with NDUFS4 OE led to improved cristae integrity and mitochondrial morphology (Fig. 4e). To correlate these results with RSCs assembly, we examined the role of NDUFS4 OE on the abundance of intact RSCs in digitonin-treated

mitochondrial samples by blue native polyacrylamide gel electrophoresis (BN-PAGE)[39,40]. The abundance of intact RSCs on BN-PAGE gels was reduced in HG conditions (Supplementary Fig. 4i, j) whereas cultured NDUFS4 OE podocytes in HG media exhibited significantly higher protein abundance of RSCs (Fig. 4f). Western blot analysis of the same samples with OXPHOS cocktail antibodies provided similar results (Supplementary Fig. 4j). Consistent with these results, the CI in-gel activity as shown by reduction of nitro-blue tetrazolium in the presence of NADH, was reduced in podocytes treated with HG as compared to those treated with NG (Supplementary Fig. 4k), whereas

**Fig. 3 | Ndufs4 overexpression improves mitochondrial morphology in the diabetic environment. a** OCRs in primary podocytes. Dotted lines denote injections of oligomycin (2 μM), FCCP (2 μM), rotenone, and antimycin A (both 0.5 μM). Basal respiration (**b**), maximal respiration (**c**), ATP-linked OCR (**d**), and spare OCR (**e**) (**a–e**, n = 6 (WT), n = 7 (*Ins2^Akita/+* and *Ndufs4^PodTg^;Ins2^Akita/+^*), n = 8 (*Ndufs4^PodTg^*), replicates/group). **f** Primary podocytes susceptibility to rotenone (left panel), and rotenone half maximal inhibition (IC_{50}) (right panel, n = 6 replicates/group). **g** Representative glomerular CI activity assessed by NADH oxidase staining from kidney sections (left panel) in different group of mice (n = 60 from 3 mice/group, scale bar = 50 μm), and quantitative analysis using Image J software normalized to the median intensity of the WT (right panel). **h** Representative podocyte mitochondria from kidney tissues (top row) and pseudo-color superimposed images (middle row) to assess mitochondrial morphological changes, including aspect ratio (**i**), circularity (**j**), roundness (**k**), and feret diameter (**l**) (n = 312 (WT), n = 167 (*Ins2^Akita/+^*), n = 141 (*Ndufs4^PodTg^*) and n = 284 (*Ndufs4^PodTg^;Ins2^Akita/+^*), measurements

from 3 mice/group). TEM micrographs from primary podocytes isolated from experimental mice (bottom row). Scale bars = 200 nm. Quantitative analyses of cristae number (**m**) and junction (**n**) normalized by mitochondrial area using TEM micrographs of primary podocytes isolated from mice (n = 45 mitochondria/group). **o** Representative images of mROS in primary podocytes assessed by mito-roGFP (Left panel). The heatmap shows the ratio of the oxidized to reduced roGFP signals in mitochondrial matrix. Quantification of mitochondrial redox states (right panel). (n = 20–22 cells/group). Scale bar = 10 μm. Total (**p**) and mitochondrial (**q**) ATP production rates in primary podocytes assessed by Seahorse Analyzer (n = 9 (WT), n = 8 (*Ndufs4^PodTg^, Ins2^Akita/+^*, and *Ndufs4^PodTg^;Ins2^Akita/+^*), replicates/group). Data are presented as mean ± SEM (**a–f, p, q**) or median ± IQR (**g, i–o**, bold line: median, and dot line: IQR). ****P < 0.0001. One-way ANOVA with post-hoc Tukey–Kramer test (**b–f, p, q**), or Kruskal–Wallis with post-hoc Dunn's test (**g, i–o**). Source data are provided as a Source Data file.

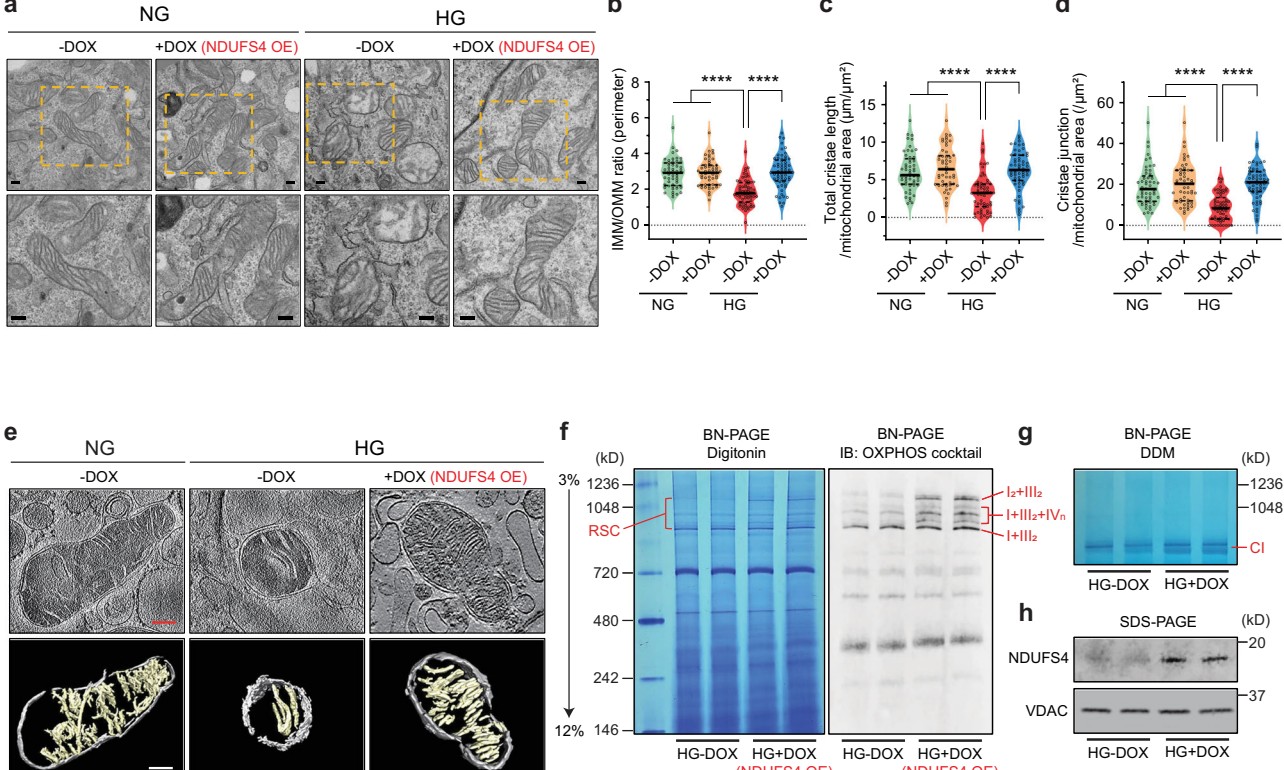

**Fig. 4 | Cristae integrity and RSCs formation are restored by Ndufs4 overexpression.** Representative TEM micrographs of cristae morphology (**a**) and quantitative analyses of cristae density (**b–d**) in podocytes cultured under NG (5.5 mM) or HG (25 mM) for 48hrs with DOX induction (NDUFS4 OE) or without DOX induction (n = 53–61 mitochondria/group, scale bars = 200 nm). IMM inner mitochondrial membrane, OMM outer mitochondrial membrane. **e** Cryo-ET analysis of purified mitochondria in DOX-inducible Ndufs4 transfected podocytes cultured under NG without DOX, HG without DOX, and HG with DOX induction (NDUFS4 OE). Slices through representative Cryo-ET tomograms are shown in upper panels and corresponding segmentations of characteristic features in lower panels (Scale bars = 200 nm). **f** Resolution of ETC complexes by BN-PAGE of mitochondria solubilized by digitonin from HG-DOX (control) or HG + DOX

(NDUFS4 OE) podocytes. Left panel: Coomassie staining; right panel: Immunoblots with OXPHOS cocktail antibodies. RSC respiratory supercomplexes. **g** Representative CI in-gel activity of n-Dodecyl β-D-maltoside (DDM)-solubilized mitochondria isolated from the same cells as in Fig. 4f. **h** Immunodetection of NDUFS4 after SDS-PAGE of mitochondria proteins in cells from Fig. 4f. VDAC was used as a loading control. Data are presented as median ±IQR (Bold line: median, and dot line: IQR). ****P < 0.0001. Kruskal–Wallis with post-hoc Dunn's test (**b–d**). Cryo-ET analysis was performed two times and confirmed the reproducibility of cristae morphological change in 5 mitochondria from each group of cells. BN-GEL and the subsequent immunoblot were conducted two times independently to validate the reproducibility. Source data are provided as a Source Data file.

NDUFS4 OE restored the CI in-gel activity under HG conditions (Fig. 4g, h). Taken together, our findings suggest that NDUFS4 OE regulates not only cristae morphology but also RSCs assembly and mitochondrial dynamics in kidney podocytes.

## NDUFS4 interaction with cristae regulatory proteins
The regulatory effect of NDUFS4 on cristae morphology, RSCs integrity and mitochondrial dynamics raises several questions regarding the

underlying protective molecular mechanism of NDUFS4 OE. It is known that cristae integrity is necessary for the proper spatial distribution of RSCs[11,41]. However, whether changes in ETC integrity could result in cristae remodeling is not well understood. We suspected that the effects of NDUFS4 on cristae remodeling could be through its interaction with one or several cristae regulatory proteins. To this end, we first assessed the abundance of some of the key cristae regulatory proteins by Western blots in podocytes from WT, *Ndufs4^PodTg^*, diabetic *Ins2^Akita/+^*, and

diabetic *Ndufs4^PodTg* mice. The cristae regulatory proteins, including STOML2 (Stomatin-like protein 2), IMMT/MIC60 (Inner membrane mitochondrial protein), ATAD3A (ATPase family AAA domain containing 3A) and OPA1 (Optic Atrophy 1 mitochondrial dynamin like GTPase) were all reduced in the podocytes of diabetic *Ins2^Akita/+* mice (Supplementary Fig. 5a). Conversely, the abundance of these proteins was significantly higher in podocytes from diabetic *Ndufs4^PodTg* (Supplementary Fig. 5a). We next adopted a proximity labeling approach to interrogate a possible interaction of NDUFS4 with cristae regulatory proteins (Fig. 5a). To this end, we engineered a podocyte cell line that stably expressed a DOX-inducible NDUFS4-APEX2 chimeric protein (Supplementary Fig. 5b). Biotin-labeled proteins in proximity to NDUFS4-APEX2 activated with $H_2O_2$ were isolated using streptavidin-coupled beads and identified by LC-MS/MS. NDUFS4-APEX2 transfected podocytes without $H_2O_2$ activation or without DOX induction were used as controls. Out of 2152 proteins identified, 357 were mitochondrial proteins, and 46 of them were significantly enriched in DOX + $H_2O_2$ podocytes (Fig. 5b, c, Supplementary Fig. 5c). Among them, we found several cristae regulatory proteins, including STOML2, IMMT/MIC60, and ATAD3A (Fig. 5c, Supplementary Fig. 5c). The close proximity of NDUFS4 to the cristae regulatory proteins was further validated by immunoblot analysis using specific primary antibodies whereby the biotinylated STOML2, ATAD3A, and IMMT/MIC60 were efficiently pulled down by streptavidin beads when treated with $H_2O_2$ (Fig. 5d). Surprisingly, OPA1 was not a closely associated protein with NDUFS4 based on both the proximity labeling and strepavidin pulldown assays. Taken together, these findings suggest that the cristae regulatory proteins could potentially form complexes with NDUFS4. However, proximity labeling assay is unable to differentiate between a direct interaction or a mere close association. Additionally, it does not specify if the association of NDUFS4 with these proteins occurs in the context of individual complexes or RSCs. Consequently, we performed complexsome profiling on enriched mitochondrial samples from HG-treated cells and HG-treated NDUFS4 OE podocytes to explore whether the cristae forming proteins interact with NDUFS4 in the context of RSCs and therefore, comigrating within RSCs (Fig. 5e). We separated ETC complexes on BN-PAGE, sliced five distinct bands representing distinct RSCs (labeled 1–5) after Coomassie blue staining, and performed mass spectrometry (LC-MS/MS). A careful analysis of each band revealed that several cristae organizing proteins, including STOML2, ATAD3A and IMMT/MIC60, comigrate with RSCs, suggesting that they are in close association with RSCs structures (Fig. 5f). To further test whether STOML2, ATAD3A and IMMT/MIC60 are integrated within RSCs, we performed immunoblotting with cocktail antibodies against OXPHOS or cristae organizing proteins (Fig. 5g). We observed that whereas STOML2 was mainly colocalized with RSCs and its abundance was markedly increased with NDUFS4 OE, ATAD3 was not significantly colocalized with RSCs, and IMMT displayed a more wide-spread distribution within and outside of RSCs (Fig. 5g). Since STOML2 comigrated with RSCs in a NDUFS4 OE-dependent manner and considering our previous proximity labeling and pulldown assays (Fig. 5b–d), we decided to further pursue its role as a link between NDUFS4 OE and cristae integrity.

To further corroborate the physical interaction of NDUFS4 with STOML2, we employed a combination of biochemical and super-resolution imaging approaches. We first performed co-immunoprecipitation (Co-IP) experiments in HEK293T cells transiently transfected with NDUFS4-FLAG construct. Among pulled down proteins with FLAG antibody, both STOML2 and NDUFS4 were detected, but not with IgG antibody (Fig. 5h). We then performed a GST affinity pulldown assay in vitro in which a GST-NDUFS4 fusion protein was incubated with cell lysates transiently overexpressing a STOML2-HA fusion protein. Immunoblotting showed that STOML2 interacted with GST-NDUFS4, but not with GST control in vitro, consistent with a potential interaction between NDUFS4 and STOML2 (Fig. 5i).

To further establish whether NDUFS4 and STOML2 are spatially in close physical proximity, we employed two super-resolution imaging approaches, the stimulated emission depletion (STED) and the stochastic optical reconstruction microscopy (STORM) providing nanoscale spatial localization at a single molecule resolution[42]. The STED super-resolution microscopy following dual immunostaining of podocytes with antibodies against NDUFS4 and STOML2 showed higher degree of colocalization between NDUFS4 and STOML2 in HG-treated NDUFS4 OE podocytes compared with podocytes cultured under HG conditions (Mander's M1 coefficient=0.45 ± 0.05 for NG-DOX, 0.25 ± 0.05 for HG-DOX and 0.55 ± 0.05 for HG + DOX; M2 coefficient=0.017 ± 0.003 for NG-DOX, 0.003 ± 0.002 for HG-DOX, and 0.021 ± 0.004 for HG + DOX) (Fig. 5j, k). Similar results were confirmed in the analysis of colocalization based on the distance (Fig. 5j, l). Additional colocalization analyses of 3D images with single molecule super-resolution imaging obtained from STORM validated the spatial interaction between NDUFS4 OE and STOML2 (Fig. 5m). Specifically, among molecules with the inter-molecular distance <500 nm (Supplementary Fig. 5d), podocytes under the HG condition exhibited a significantly longer distance between NDUFS4 and the nearest neighboring STOML2, with a median nearest neighboring distance (NND) that was also significantly longer compared to podocytes under the NG or HG-treated NDUFS4 OE conditions (% colocalization = 38.9 ± 1.3% for HG-DOX *vs.* 49.8 ± 1.8% for NG and 49.5 ± 1.8% for HG + DOX; NND = 65.3 ± 3.6 nm for HG-DOX *vs.* 40.3 ± 2.9 nm for NG and 41.6 ± 3.0 nm for HG + DOX) (Fig. 5n, o). These findings suggest that within each mitochondrion, NDUFS4 and STOML2 have the highest proximity in NG or with NDUFS4 overexpression in HG conditions. In contrast, they exhibit the least spatial proximity in HG conditions.

## STOML2 is required for NDUFS4-mediated RSCs assembly and cristae integrity

To investigate the functional impact of the interaction between STOML2 and NDUFS4 on RSC remodeling, we used STOML2 knockout (KO) podocytes in the presence or absence of NDUFS4 OE. After verifying deletion of STOML2 (Fig. 6a), we performed BN-PAGE analysis followed by immunoblotting (Fig. 6b). Both Coomassie staining and immunoblotting using OXPHOS cocktail antibodies showed an increase of RSCs abundance in NDUFS4 OE podocytes. However, this increase was markedly reduced in STOML2 KO cells despite NDUFS4 OE (Fig. 6b). We also observed that HG reduced the mitochondrial cristae density in podocytes, but NDUFS4 OE restored the cristae structure and abundance in HG media and improved mitochondrial morphology (Fig. 6c–f). In the NDUFS4 OE podocytes in which the STOML2 was deleted, however, the cristae restoration was no longer observed and mitochondrial morphology was distorted in HG conditions (Fig. 6c–f). Thus, these data suggest that STOML2 is an essential component of the NDUFS4 OE-mediated improvement in cristae integrity, RSC assembly, and mitochondrial morphology.

A possible interaction between STOML2 and NDUFS4 hinges on the accessibility of Ndufs4 within RSC to STOML2. We used the UCSF ChimeraX (https://www.cgl.ucsf.edu/ chimerax/)[43], a powerful molecular modeling engine, to visualize the structure of Ndufs4 within RSC ($I_1III_2IV_1$; PDB-5XTH). This interactive molecular visualization program illustrates that CIII and CIV bind to the hydrophobic membrane arm of CI within the RSC (Supplementary Fig. 6a). Notably, it reveals that the NDUFS4 protein (magenta, Supplementary Fig. 6a) is localized on the surface of the matrix arm of the CI within the RSC, exposing NDUFS4 to potential interactions with other molecules such as STOML2.

To further address how NDUFS4 interacts with STOML2, we created a series of STOML2 deletions. The full-length mouse STOML2 protein is 353 amino acids long and contains four functional domains: the N-terminal mitochondrial-targeting sequence (MTS), a hydrophobic hairpin (HP) domain, a conserved stomatin (STOM)

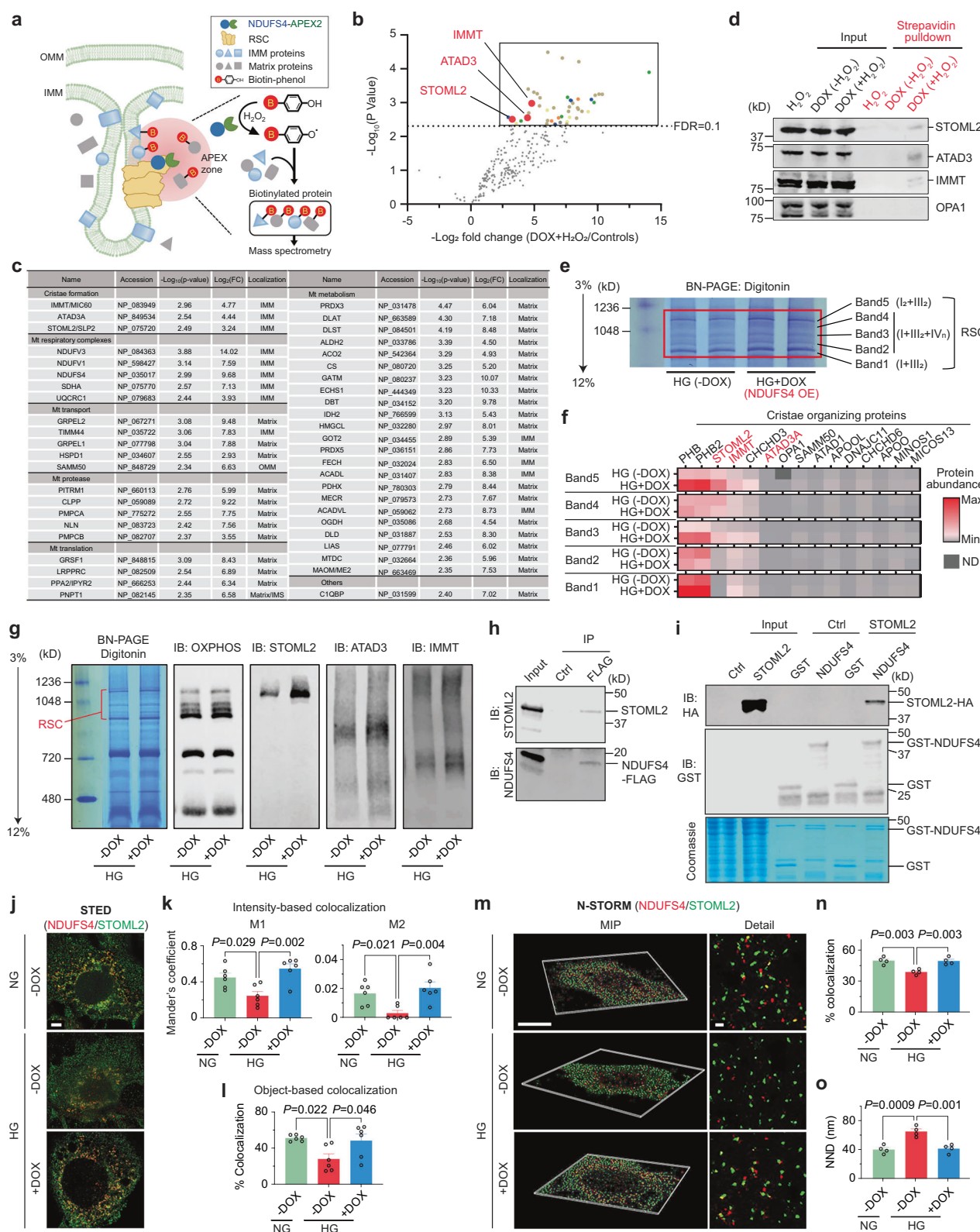

domain, and the C-terminal coiled-coil domain (CTD) (Fig. 6g). We observed that the C-terminal deletion mutant of STOML2 was co-immunoprecipitated with FLAG tagged-NDUFS4, but not with other mutants at the N-terminal domain which included HP and STOM domains (Supplementary Fig. 6b, c), suggesting that the N-terminal domain of STOML2 is the key domain for NDUFS4 binding. We next generated additional mutants harboring the N-terminal domain and performed GST-NDUFS4 pull-down assay (Fig. 6g, h). While deletion

mutants of C-terminal (ΔCTD (Δ213–353)), HP domain (ΔHP(Δ29–78)), and the α1–4 helices of the STOM domain (Δ110–168) did not prevent binding to NDUFS4, complete deletion of STOM domain (ΔSTOM (Δ79–212)), as well as deletions of β1–3 strands (Δ70–109) and β4 strand plus α5 helix (Δ169–212) of STOM domain prevented binding of STOML2 to GST-NDUFS4 (Fig. 6g, h), indicating that NDUFS4 could bind to STOML2 at the regions of β1–4 strands and α5 helix in the STOM domain. This result is

**Fig. 5 | STOML2 interacts with NDUFS4. a** Schematic depiction of APEX2 proximity labeling followed by the LC-MS/MS analysis in podocytes transduced with a DOX-inducible NDUFS4-APEX2 chimeric construct. B biotin, IMM inner mitochondrial membrane, OMM outer mitochondrial membrane, RSC respiratory supercomplexes. **b** Volcano plots of NDUFS4-associated mitochondrial proteins identified by LC-MS/MS analysis. The average values of NDUFS4-APEX2 transfected podocytes without $H_2O_2$ and those without DOX induction are used as controls. Mitochondrial proteins significantly changing (FDR Q < 0.1) are depicted in colors. The dotted line indicates a cut off for significant changes based on FDR of 0.1. **c** NDUFS4 interactomes identified by the LC-MS/MS following APEX2 labeling. **d** Immunoblotting of STOML2, ATAD3, IMMT, and OPA1 in whole cell lysates (Input) and streptavidin pulldown samples. OPA1 was used as a negative control. **e** RSCs were excised (Band 1–5) from BN-PAGE, followed by LC/MS-MS analysis. **f** Heatmap of cristae organizing proteins in the RSCs from podocytes cultured in HG with DOX induction (HG + DOX) or without (HG-DOX). **g** Immunoblotting of OXPHOS, STOML2, ATAD3 and IMMT following BN-PAGE analysis as described in Fig. 5e. **h** Coimmunoprecipitation (Co-IP) of NDUFS4 and STOML2. HEK 293T cells were transiently transfected with Ndufs4-FLAG expression construct. Input was used as a positive control while IgG was used as a negative control (Ctrl), and anti-FLAG antibody was used for Co-IP. **i** GST affinity pulldown assay in HEK293T cells overexpressing a STOML2-HA fusion protein. **j** Representative images of NDUFS4 (red) and STOML2 (green) by STED analysis in cultured podocytes under the NG, HG, and HG plus NDUFS4 OE conditions (Scale bar = 5 μm). **k** Intensity-based colocalization expressed by Mander's coefficient. M1 and M2 indicate the coefficient for NDUFS4 overlapping STOML2 and STOML2 overlapping NDUFS4, respectively (n = 6 cells). M2 coefficient was substantially lower than M1 since STOML2 as compared to NDUFS4 is more widely distributed both in cytoplasm and mitochondria[77]. **l** Object-based colocalization expressed by percent colocalization (n = 6 cells). **m** Representative maximal intensity projection (MIP) images and detailed molecular images of NDUFS4 (red) and STOML2 (green) by STORM in cultured podocytes under the same conditions as shown in Fig. 5j. Scale bars: 10 μm (left panel) and 500 nm (right panel). **n** Percent colocalization based on the nearest neighboring distance (NND) < 40 nm (n = 4 cells). **o** Median NND (n = 4 cells). Data are presented as mean ± SEM. One-way ANOVA with post-hoc Tukey–Kramer test (**k, l, n, o**). Immunoblot experiments were performed two times independently to validate the reproducibility. Source data are provided as a Source Data file.

consistent with a molecular docking simulation analysis which showed two regions, 70–109 and 169–212, characterized by four β strands in STOM domain (Fig. 6g), have a high probability to interact with NDUFS4 within the RSC structure[44] (Supplementary Fig. 6a). Further deletions of smaller and more defined secondary structures of STOML2 revealed that while β1 and α5 are dispensable for STOML2 interaction with NDUFS4, deletion of β2 (11 amino acids), β3 (12 amino acids), or β4 (13 amino acids) strand significantly impair this interaction (Fig. 6g, i). Consistent with these findings, the molecular docking analysis (Fig. 6g) also suggests that β2–4 strands of STOML2 create a structural interphase in the conserved STOM domain that facilitates its binding to NDUFS4.

## Discussion

The findings of this study advance our current understanding of the pathobiology of mitochondrial respiration and its central role in the pathogenesis of DKD and provide a mechanistic link between NDUFS4 deficiency in podocytes and DKD progression in vivo (Fig. 7). Using several independent experimental models of DKD, we uncover the central role of ETC integrity as a defining feature of mitochondrial dysfunction and a powerful regulator of cristae remodeling and mitochondrial dynamics in the diabetic milieu. Importantly, our study provides strong experimental data to support an interesting biological function of Ndufs4 in preserving the integrity of cristae morphology and suggests that Ndufs4 ties CI integrity and maintaining the structural integrity of cristae in podocytes to high glucose metabolic cues in the cell.

A first hint that NDUFS4 deficiency could have a key role in defining mitochondrial dysfunction in DKD came from our initial comparative proteomic profiling revealing a consistent downregulation of several subunits of CI in diabetic podocytes. We reasoned that one possible explanation for the CI remodeling in podocytes could be an initial adaptive mechanism to the diabetic environment. However, as hyperglycemia persists, these chronic changes might have become maladaptive and pathogenic in nature resulting in biochemical and structural alterations in mitochondria. To this end, we argued that forced expression of the NDUFS4 subunit, consistently downregulated in the podocytes of the type 1 and type 2 diabetic mice as well as in the glomeruli of the DKD patients, might overcome the mitochondrial maladaptation in DKD and provide significant insights into molecular mechanisms of its progression. Consistent with our hypothesis, we found that overexpression of the NDUFS4 in podocytes improved mitochondrial respiration and CI activity and prevented mitochondrial fragmentation in podocytes of diabetic mice. We also linked NDUFS4 overexpression with the protection of both RSC assembly and cristae morphology.

Our qualitative and quantitative approaches in this study, including tomography, clearly suggest an aberrant cristae morphology in the diabetic milieu. Remarkably, we find that these alterations in cristae structure were markedly improved with the forced expression of Ndufs4. While previous studies have identified the critical role of cristae shaping proteins, including STOML2, in cristae remodeling and RSCs assembly[41,45,46], the interaction between NDUFS4 and cristae shaping proteins remained largely unknown. Our findings suggest that the interaction between NDUFS4 and STOML2 is necessary for the proper maintenance of cristae morphology, RSC integrity, and ETC function in podocytes. Our data suggests that reduced levels of NDUFS4 in diabetic conditions result in compromised interactions between NDUFS4 and STOML2. The reduced interaction between STOML2 and NDUFS4 leads to a disorganized cristae platform for the assembly of RSCs and contributing to decreased CI function and mitochondrial dysfunction. Overexpressing Ndufs4 in diabetic podocytes improves this interaction leading to enhanced CI stability, improved mitochondrial RSCs assembly and cristae morphology. Furthermore, we validated this interaction using a variety of biochemical and imaging techniques, including proximity labeling, Co-IP, GST pulldown assay, complexsome profiling, and super-resolution imaging approaches. An extensive domain mapping analysis identified three crucial β strands of STOML2 that serve as binding sites for NDUFS4. Furthermore, the molecular docking analysis suggests that these adjacent β strands fold into a unique antiparallel β-pleated sheet in the conserved STOM domain that facilitates the binding of STOML2 to NDUFS4. Taken together, our findings suggest that the interaction between NDUFS4 and STOML2 could be an important mechanism adapting the CI assembly and function in response to the metabolic cues of the cell.

Our study has unraveled many unexpected aspects of pathobiology of NDUFS4 in podocytes and its role in the progression of DKD, however, our findings also raise several important questions that remain to be fully addressed. For example, what upstream signaling pathways are required to initiate the cascade of events that lead to reduced expression of NDUFS4 in podocytes in DKD? Furthermore, additional experiments are needed to determine the extent of NDUFS4 deficiency in other cells and tissues. It is important to emphasize that while our results support the critical role of Ndufs4 on mitochondrial function and DKD progression, this does not imply exclusivity. Further studies and a comprehensive exploration of the impact of other subunits of CI on progression of DKD are needed to better define the role of Ndufs4 on progression of DKD and other kidney diseases. The interplay between NDUFS4 and STOML2 is also complex and further studies are needed to test whether STOML2 can be regarded as a specific and necessary binding partner of NDUFS4 in maintaining

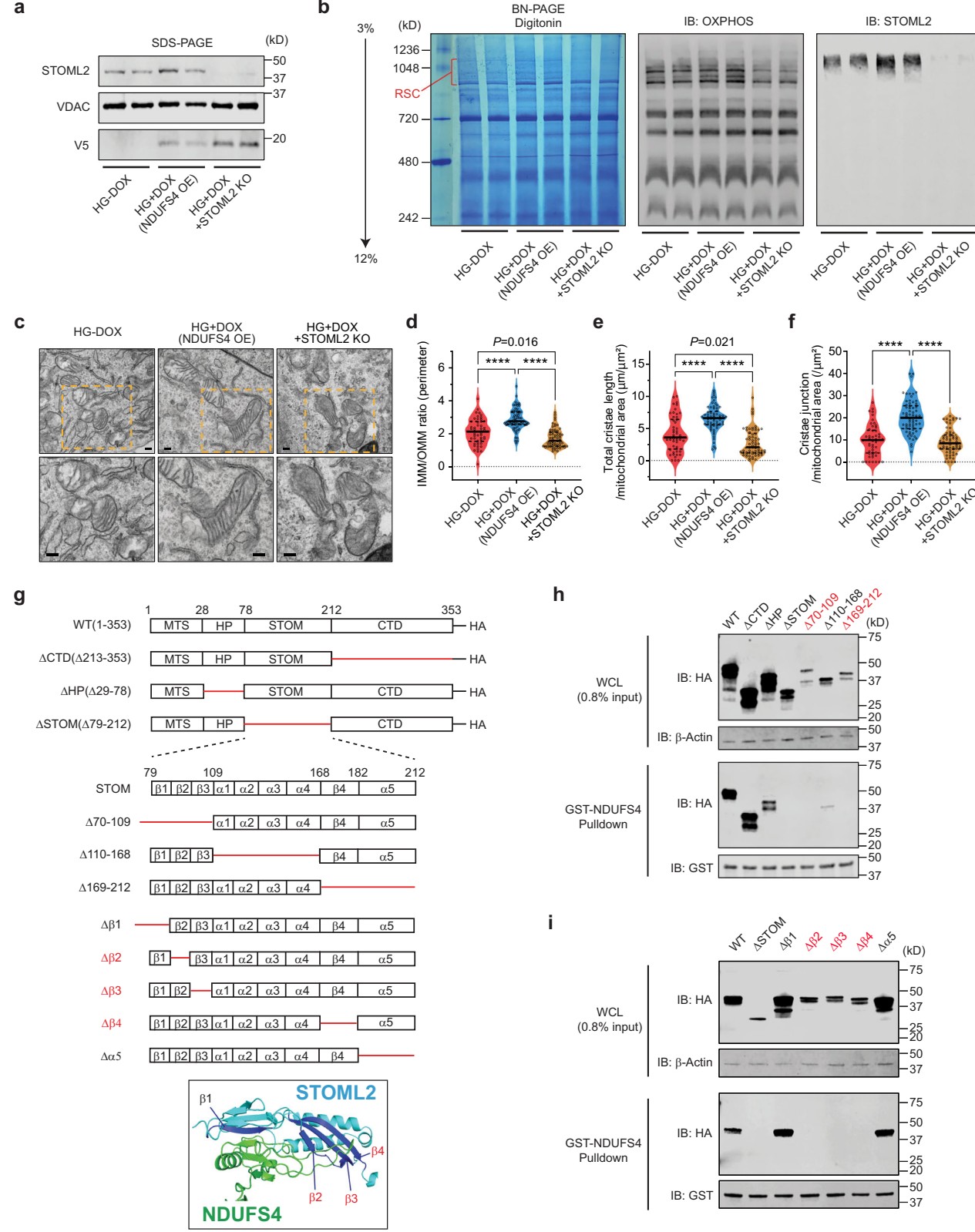

cristae structure. Finally, one important question is whether our findings are relevant to humans. Our data indicates that reduced levels of glomerular NDUFS4 expression correlate with albuminuria and eGFR in subjects with DKD. Importantly, we find that NDUFS4 staining in glomeruli is progressively reduced with worsening of DKD histology suggesting that Ndufs4 may play an important role in progression of DKD. However, mice and humans present major differences that might influence the dynamics of the events described in this study and further studies are required to validate our results in individuals with DKD.

In summary, our study suggests that reduced levels of NDUFS4 expression leads to compromised CI and RSCs formation with a significant effect on bioenergetic capacity, cristae integrity, and mitochondrial morphology of podocytes promoting DKD progression. We

**Fig. 6 | STOML2 is essential for NDUFS4-mediated improvement of RSCs formation and cristae remodeling. a** Immunoblot of STOML2 in DOX-inducible NDUFS4-V5 podocytes transduced with lentiviral CRISPR with nontargeting sgRNA control (HG + DOX) or with STOML2 targeted sgRNA (HG + DOX + STOML2KO); podocytes without Ndufs4 induction were used as control (HG-DOX). NDUFS4 was blotted with V5 tag antibody. **b** BN-PAGE of digitonin-solubilized mitochondrial proteins from the same cells as shown in Fig. 6a stained with Coomassie or immunoblotted (IB) with OXPHOS cocktail or STOML2 antibodies. RSC respiratory supercomplexes, DOX doxycycline. **c** Representative TEM micrographs of mitochondria in cells described in Fig. 6a, and cristae density measurements by IMM/OMM ratio (**d**), total cristae length (**e**) and cristae junction (**f**), (*n* = 50–65 mitochondria/group, scale bars = 200 nm). IMM inner mitochondrial membrane, OMM outer mitochondrial membrane. **g** Structure of mouse STOML2 wild type (WT) and deletion mutant constructs engineered with HA-tag at the C-terminus. MTS

mitochondrial targeting sequence, HP hydrophobic hairpin, STOM stomatin, CTD C-terminal coiled-coil domain, β1–4 β1–4 strands in STOM domain, α1–5 α1–5 helices in STOM domain. Ribbon diagram in the box represents a protein-protein docking simulation of human STOML2 (blue) and NDUFS4 (green; in the context of human RSC). **h, i** GST pull-down assay from HEK293T cells overexpressing STOML2-HA fusion protein or deletion mutants. WT full length STOML2, ΔCTD deletion mutant of C-terminal domain, ΔHP deletion mutant of hydrophobic hairpin domain (Δ29–78), ΔSTOM deletion mutant of stomatin domain, Δ70–109 β1–3 deletion mutant, Δ110–168 α1–4 deletion mutant, Δ169–212 β4 strand and α5 helix deletion mutant (**h**). Δβ1–Δβ4 deletion mutants of individual β strand, Δα5 α5 deletion mutant (**i**). Data are presented as median ± IQR (Bold line: median, and dot line: IQR). ****P < 0.0001. Kruskal–Wallis with post-hoc Dunn's test (**d**–**f**). Immunoblot experiments were performed two times independently to validate the reproducibility. Source data are provided as a Source Data file.

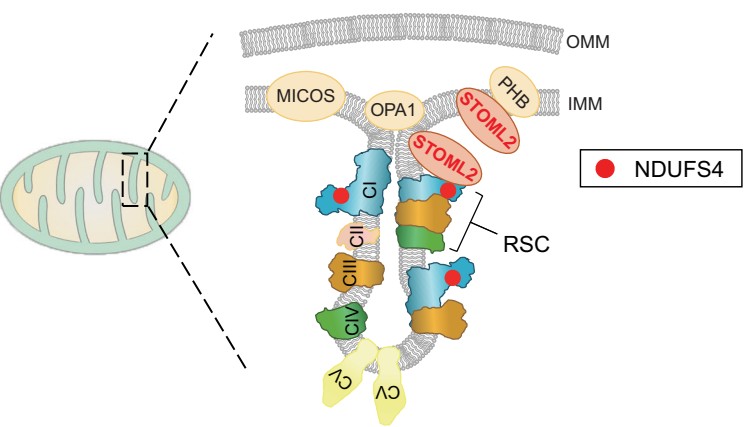

## Mitochondrial crista structure

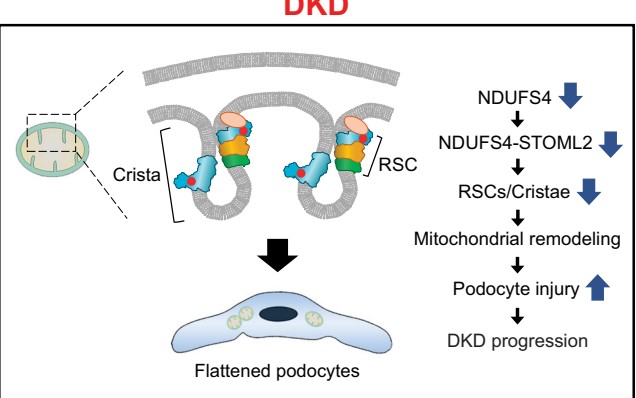

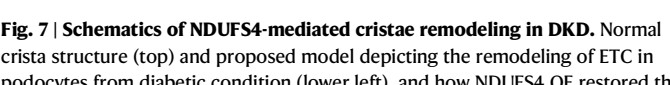

**Fig. 7 | Schematics of NDUFS4-mediated cristae remodeling in DKD.** Normal crista structure (top) and proposed model depicting the remodeling of ETC in podocytes from diabetic condition (lower left), and how NDUFS4 OE restored the mitochondrial morphology and function (lower right). DKD diabetic kidney disease, IMM inner mitochondrial membrane, OMM outer mitochondrial membrane, RSC respiratory supercomplexes.

discovered that forced expression of NDUFS4 in podocytes under the diabetic environment, however, leads to significant improvement in RSCs assembly and cristae and mitochondrial morphology mitigating DKD progression. We propose that strategies aimed at improving NDUFS4 expression in DKD could emerge as a paradigm shifting intervention for ameliorating progression of DKD.

## Methods
### Mouse models
All animal studies were reviewed and approved by the Institutional Animal Care and Use Committee (IACUC) of the University of Texas at

MD Anderson Cancer Center (MDACC) and conducted according to the institutional and the US National Institutes of Health guidelines. All mice were maintained in a temperature-controlled environment (22°C) and 50-60% humidity under a 12-hrs light/dark cycle with *ad Libitum* feeding (Picolab Rodent Diet 20; product code#5053). Type 1 and type 2 diabetic mice (*Ins2^Akita/+* on C57BL/6J background and *Lepr^db/db* on C57BLKS/J background) were obtained from Jackson Laboratories (Stock Nos. 003548 and 000642). Only male mice were used in this study as the males in these mouse models develop more severe diabetic kidney disease. Urine samples were used to measure albumin and creatinine concentration using mouse albumin ELISA kit (Ethos

Biosciences) and QuantiChrom creatinine assay kit (Bioassay Systems), respectively. Hemoglobin A1c was measured by HbA1c Kit (Crystal Chem). The Podocyte-specific Ndufs4 transgenic mice were generated using a podocyte-specific transgenic Ndufs4 construct under the control of a human podocin promoter driving a murine Ndufs4 cDNA followed by a WPRE and a human growth hormone polyadenylation signal. The transgenic cassette was flanked by chicken hypersensitivity site 4 (HS4) insulators. The DNA construct was restriction digested, purified and microinjected into pronuclei of C57BL/6J embryos by the BCM Genetically Engineered Rodent Models Core. Transgenic mice genotyping was performed by PCR using genomic DNA isolated from mouse tails with the following primers: 5′-ACTCCA-CAGGGACTGCGCTC-3′; 5′-CCGAGTCTGGTTGTCTGCCA-3′. A 217-bp DNA fragment can be amplified only from the Ndufs4 transgenic mice. Following the IACUC guidelines of MDACC, we used the carbon dioxide (CO$_2$) inhalation method for the euthanasia of mice used in this study.

### Cell lines
Conditionally immortalized mouse podocytes were a kind gift from Jochen Reiser (Rush University, Chicago, IL). Cells were cultured at 33 °C in RPMI (Corning) containing 10% FBS (GenDepot), antibiotic antimycotic solution (Corning), and 20 U/ml mouse recombinant IFN-γ (Sigma) to enhance expression of a thermosensitive T antigen. The cells were differentiated at 37 °C in DMEM (Corning) supplemented with 5% FBS and antibiotic antimycotic solution without IFN-γ on collagen type I (Gibco) coated dishes for 7–12 days. For imaging, differentiated cells were trypsinized, dissociated, and plated onto collagen type I coated coverslips. Podocytes prepared for experiments involving high glucose conditions (HG, 25 mM) were serum deprived for 24 hours prior to addition of HG. Control cells were cultured with normal glucose (NG, 5.5 mM). For primary cultured podocytes, isolated podocytes from WT and *Ndufs4*$^{PodTg}$ mice were cultured with DMEM containing NG, 5% FBS, and antibiotic antimycotic solution, while those from *Ins2*$^{Akita/+}$ and *Ins2*$^{Akita/+}$;*Ndufs4*$^{PodTg}$ mice were cultured with DMEM containing HG, 5% FBS, and antibiotic antimycotic solution. HEK 293 T cells (ATCC CRL-3216) were cultured in high glucose DMEM with 10% FBS and antibiotic antimycotic solution.

### Cloning and genome editing
Mouse Ndufs4 cDNA fused with a 3′-V5 tag was engineered by PCR, then subcloned into a *PiggyBac* transposon-based vector with TRE promoter and hygromycin selection marker[47]. Stoml2 cDNA (NM_023231) was purchased from OriGene. Stoml2 deletion mutants were generated using Q5 Site-Directed Mutagenesis Kit (New England Biolabs) and verified by sequencing. Gene editing of Stoml2 was carried out using CRISPR Cas9[48] in an engineered *PiggyBac* system with puromycin selection. Two CRISPR gRNAs targeting exon 1 and exon 2 of Stoml2 are: GTGGGAAATGCTGGCGCGCG<u>CGG</u> and TCACCGGTTCCAGGATCCGG<u>TGG</u>.

### Human kidney biopsy samples
Human kidney biopsy samples were obtained from Okayama University Hospital in Okayama, Japan. The protocol was approved by the Institutional Review Board of the Ethics Committee of Okayama University Hospital and registered with the University Hospital Medical Information Network (UMIN) (identification number: UMIN000046398). Written informed consent was obtained from all participants in this study. For the control group, we analyzed kidney biopsy samples at the time of kidney transplantation from 9 donor participants. DKD was classified and histological scores were determined according to the criteria of the Renal Pathology Society[31]. For immunohistochemistry, antigen retrieval was performed by heating sections in 10 mM citrate buffer (pH 6.0) in a microwave oven. Reactions with endogenous peroxidases and proteins were blocked by

incubation with 0.3% H$_2$O$_2$ diluted in methanol and serum-free protein blocking solution (Dako). Then, tissue was incubated with rabbit anti-NDUFS4 primary antibody (Abcam, ab137064) overnight at 4 °C. Goat anti-rabbit IgG HRP Polymer (Vector Laboratories, MP-7451) was used as the secondary antibody and peroxidase activity was visualized with a liquid diaminobenzidine substrate (Vector Laboratories). Image J was used for the quantification of glomerular NDUFS4 intensity. Briefly, the mean intensity of NDUFS4 in glomerular tuft area (pixel/μm$^2$) was determined after subtracting the background intensity as described previously[49]. Gene expression data were extracted from Nephroseq database version 5 (https://www.nephroseq.org). Glomerular transcriptomic data were analyzed using Ju CKD Glom dataset.

### Doxycycline (DOX)-inducible transient Ndufs4 expression
A DOX-inducible Ndufs4 overexpression construct, based on a reverse tetracycline-controlled transactivator (rtTA) and tetracycline-responsive element promoter (TRE), was engineered in the *PiggyBac* transposon system to overexpress Ndufs4 in the Tet-ON system (Supplementary Fig. 4a). Doxycycline (Sigma) was added to the culture media at a concentration of 200 nM for 72 h, or as indicated in the figures. After induction, cells were harvested and lysed in 1× Laemmli Sample Buffer (BioRad) and proceed to SDS-PAGE and immunoblotting analysis for Ndufs4 expression.

### RNA extraction and real time qRT-PCR
Cells were homogenized in TRIZOL (Invitrogen), and total RNA was purified using PureLink™ RNA Mini Kit (Invitrogen) according to the manufacture's protocols. Real-time quantitative RT-PCR (qRT-PCR) was performed using the SYBR green dye (Applied Biosystems) with a StepOnePlus Real-Time PCR System (Applied Biosystems). Fold changes of gene expression was normalized by housekeeping genes and analyzed using the ΔΔCT method. The specific primers for target gene used in this study are listed in Supplementary Table 2.

### SDS-PAGE
Western blot assays were performed as described previously[13]. In brief, cells or purified mitochondria were resuspended in RIPA buffer (TEKnova) containing 1% protease inhibitor cocktail (Sigma). Protein concentration was determined using BCA protein assay (Pierce). To analyze mitochondrial proteins, lysates were heated at 42 °C for 5 min in Laemmli sample buffer (Bio-Rad), but proteins from whole cell lysate were heated at 95 °C for 5 min in the same buffer. 10–30 μg protein lysates were loaded onto 4–20% gradient SDS PAGE (Bio-Rad) and transferred to PVDF membranes (Roche). Membranes were probed with the primary antibodies followed by washing and adding the fluorescent secondary antibodies. Antibody-antigen reaction profiles were visualized and quantified by Odyssey XF Imager (LI-COR). Antibodies are summarized in Supplementary Table 3.

### Podocyte and tubular cell isolation
Podocyte and tubular cell isolation from mouse kidneys were performed as previously described with slight modifications[50,51]. In brief, podocytes were ex vivo selected by biotin-labeled anti-Kirrel3 and podocalyxin antibodies (R&D Systems BAF4910 and R&D Systems BAF1556, respectively), then further purified using magnetic, streptavidin conjugated Dynabeads (Thermo Fisher). To isolate tubular cells, dissected and minced kidneys were digested with collagenase type II in RPMI media for 30 min at 37 °C. Cells were sieved first through a 100 μm nylon mesh, then through a 40 μm nylon mesh, followed by centrifugation at 500 g for 10 min. The pellet was resuspended in red blood cell lysis buffer (R&D Systems) and incubated on ice for 10 min. After centrifuge at 500 g for 10 min, cells were resuspended in RIPA buffer (TEKnova) containing 1% protease inhibitor cocktail (Sigma) and stored at -80°C freezer for experiments.

## Mitochondrial isolation

Mitochondrial isolation was performed using Percoll density gradient centrifugation[52]. Mouse primary podocytes were resuspended in mitochondrial isolation buffer (MIB1: 10 mM HEPES, 250 mM Sucrose, and 1 mM EDTA, pH 7.4, at 4 °C), and homogenized by a glass homogenizer, followed by centrifuging homogenate at $1300 \times g$ at 4 °C for 3 min. After two cycles of homogenization and centrifugation, the pooled supernatant was centrifuged at $21,000 \times g$ at 4 °C for 10 min. The resultant pellet was resuspended with 15% Percoll in MIB1 followed by purification by Percoll density centrifugation using a stepwise density gradient of 40%, 23%, and 15% Percoll in MIB1, at $30,700 \times g$ at 4 °C for 5 min. Mitochondria accumulating at the interface between the 23% and 40% were collected, washed with MIB1, and centrifuged at $16,800 \times g$ at 4 °C for 10 min. Mitochondrial pellet was resuspended with MIB1 and centrifuged at $7000 \times g$ at 4 °C for 10 min. Purified mitochondria was resuspend with MIB1, snap-frozen in liquid nitrogen, and stored at −80 °C freezer for experiments. Mitochondria from cultured podocytes were isolated using Sucrose step density gradient centrifugation method with some modifications[53]. In brief, cells were resuspended in another mitochondrial isolation buffer (MIB2: 3 mM HEPES, 210 mM Mannitol, 70 mM Sucrose, and 0.5 mM EDTA, 1 mM MgCl2, pH 7.4, at 4 °C), and homogenized using a syringe with a 27.5 G needle. Homogenate was centrifuged at $500 \times g$ at 4 °C for 5 min, and the resultant pellet was homogenized with MIB2 by a glass homogenizer, followed by the centrifugation at $500 \times g$ at 4 °C for 5 min. After the supernatants were combined, the pooled supernatant was carefully added on the top of the same amount of 340 mM sucrose in a tube and centrifuged $500 \times g$ at 4 °C for 10 min with low acceleration/deceleration. After the centrifugation, the top layer (mitochondrial fraction) was transfer to the fresh tube and centrifuged at $10,000 \times g$ at 4 °C for 10 min. Pellet was resuspended with MIB2 and centrifuge at $7000 \times g$ at 4 °C for 10 min. Purified mitochondria was resuspend with 300 mM sucrose, snap froze in liquid nitrogen, and stored at −80 °C freezer for experiments.

## Mitochondrial OCR, ECAR, RCR, Energy map, and the real-time ATP rate assays

Mitochondrial OCR, ECAR, and the real-time ATP production rate were measured in cultured primary podocytes using a Seahorse Bioscience XFe-96 Analyzer according to the manufacturer's instructions (Agilent Technologies). The optimal cell density for podocytes and the concentration of different drugs were previously described[50]. In brief, primary podocytes were seeded on 96-well culture plates coated with collagen I (0.1 μg/ml), and the following drugs were injected: oligomycin (2 μM), FCCP (2 μM), rotenone (0.5 μM), and antimycin A (0.5 μM) for the mitochondrial respiratory assay, and oligomycin (1.5 μM), rotenone (0.5 μM), and antimycin A (0.5 μM) for the real-time ATP rate assay. Data were normalized by cell numbers using the CyQUANT assay kit (Invitrogen) and analyzed using Wave (version 2.6.0, Agilent) and Prism software packages (version 9, Graphpad). The respiratory control ratio (RCR) was calculated as the ratio between the uncoupler-stimulated respiration and the proton leak-linked respiration[54]. Energy maps were generated using the basal and maximum values for OCRs and ECARs in the mito-stress test with the above-mentioned ETC modulators' conditions. For the evaluation of rotenone sensitivity, podocytes were treated with different concentrations of rotenone (10 nM, 50 nM, 100 nM, and 250 nM) after stimulating with FCCP (2 μM). In each sample, dose-response curve was individually developed and the rotenone IC$_{50}$ value was calculated to compare the sensitivity to CI inhibition.

## Measurement of ATP content

Total ATP content was assessed by CellTiter-Glo 2.0 Cell Viability Assay Kit (Promega) according to the manufacturer's instructions. ATP measurements were normalized by DNA concentration measured by CyQUANT Cell proliferation Assay Kit (Molecular Probes). In brief, cells were cultured in 100 μl/well media on 96 well Black/Clear Bottom Collagen Plate (Thermo Fisher) followed by glucose and/or DOX treatment. Cells were equilibrated to room temperature for 30 min before addition of 100 μl premade CellTiter-Glo 2.0 Reagent and incubated for 10 min followed by luminescence recording on a FLOUstar Omega microplate reader (BMG LABTECH). The same plate was then washed with PBS twice and added 200 μl/well of CyQuant-GR diluted in 1× Cell Lysis Buffer for 5 min incubation. Fluorescence readings from the bottom using 480 nm excitation and 520 nm emission were converted to cell numbers using a standard curve.

## Mitochondrial roGFP2

The redox sensitive mitochondrial matrix-targeted roGFP2 (mito-roGFP2) was assessed as previously described with some modifications[55]. Briefly, the mito-roGFP2 construct was engineered by ligating a 48-bp nucleotides DNA encoding the mitochondrial targeting sequence from the cytochrome oxidase subunit IV to the 5' of the roGFP2 coding sequence[33]. This fusion cDNA was cloned into lentiviral vector LVX-EF1-GFP2 (Addgene 134867) by replacing the GFP cDNA between SpeI and BsrGI. Primary podocytes were transiently transduced with the lentiviral-mito-roGFP2 and cultured in experimental conditions before live cell imaging analysis. Cells were imaged by laser scanning confocal microscopy (Leica SP8) using two excitations, 405 nm and 488 nm to preferentially excite oxidized roGFP and reduced roGFP, respectively. Images were quantitatively analyzed using Imaris (Andor/Oxford Instruments, version 9.9) to determine mitochondrial redox ratio according to the mean fluorescence intensity (MFI) per channel.

## MitoSOX measurement

Cells were cultured at 37 °C and treated with glucose and/or DOX for 72 h. Cells were then washed twice with HBSS and treated with fresh 5 μM MitoSOX Red (Thermo Fisher) in HBSS media for 15 min. Cells were washed again with HBSS for 3 times, trypsinized and resuspended in 0.2 ml PBS for FACS analysis using 396 nm excitation and 510 nm emission[13].

## Blue-Native PAGE

Mitochondrial enriched fractions were centrifuged at $10,000 \times g$ at 4 °C for 5 min. The mitochondrial complexes were then solubilized in digitonin (8 μg/μg mitochondrial protein, Sigma) or DDM (3 μg/μg mitochondrial protein, Thermo Fisher) with 4X NativePAGE sample buffer (Thermo Fisher). Following centrifugation at $20,000 \times g$ for 10 min at 4 °C to remove insoluble material, Coomassie G-250 was added to achieve a final concentration one-fourth of the detergent concentration. Next, 15 μg of proteins were run on a pre-cast 3–12% gradient Native PAGE Bis-Tris Gel (Invitrogen) at 150 V at 4 °C for 30 min with Cathode Buffer B (50 mM Tricine, 7.5 mM imidazole, 0.02% Coomassie G-250, pH 7.0), followed by 2.5 h at 250 V at 4 °C with 1/10 Cathode Buffer B (50 mM Tricine, 7.5 mM imidazole, 0.002% Coomassie G-250, pH 7.0)[39,56]. Lastly, BN-PAGE was followed by Coomassie staining and destaining, or transferring to PVDF membrane for immunoblotting analysis.

## Complex I activity

Rotenone-sensitive complexes I activity was assayed spectrophotometrically by measuring the decrease in absorbance at 340 nm resulting from the oxidation of NADH in isolated mitochondria according to established methods[57–59]. In brief, mitochondria (10–20 μg protein) were freeze-thawed to disrupt the membranes before added to assay buffer (25 mM KH2PO4/K2HPO4 and 5 mM MgCl2, pH 7.2) with 2 mM KCN, 2.5 mg/mL BSA, 2 μg/mL antimycin A, and 0.13 mM NADH with or without 2 μg/mL rotenone. The assays were initiated by the addition of 65 μM ubiquinone as electron acceptor and

the kinetics of absorbance at 340 nm was measured on a FLOUstar Omega microplate reader (BMG LABTECH). CI activity was also measured with a CI Enzyme Activity Microplate Assay Kit (Abcam) using 10–20 µg of isolated mitochondria according to the manufacturer's instructions. CI activity in kidney tissue sections was assessed by NADH oxidase staining based on a previous report[32]. NADH oxidase activity was assayed by incubating kidney sections in 50 mM Tris-HCl (pH 7.4), 0.8 mg/ml NADH (Sigma), and 1 mg/ml nitro blue tetrazolium (NBT, Sigma) for 1 h at RT. After washing in distilled water three times, samples were washed with 3 exchanges of the 30%, 60%, 90% acetone solutions in increasing then decreasing concentration to remove unbound NBT. After a rinse in distilled water, slides were mounted with the aqueous mounting medium. The intensity of CI activity was measured using image J. For each glomerulus, we determined the intensity divided by the median intensity observed in the WT (wild type) images. To assess in-gel activity for CI, BN-PAGE gels were incubated in the assay buffer consisting of 5 mM Tris-HCl (pH 7.4) with 0.1 mg/ml NADH (Sigma) and 2.5 mg/ml NBT (Sigma) for 15 min at RT as described previously[39].

## Scanning electron microscopy

Scanning electron microscopy was conducted as previously reported[14]. In brief, tissue samples fixed in solutions containing 3% glutaraldehyde plus 2% paraformaldehyde in 0.1 M cacodylate buffer (pH 7.3) were washed with 0.1 M cacodylate buffer (pH 7.3), postfixed with 1% cacodylate buffered osmium tetroxide (OsO4), washed with 0.1 M cacodylate buffer, then in distilled water. Afterwards, the samples were sequentially treated with Millipore-filtered 1% aqueous tannic acid, washed in distilled water, treated with Millipore-filtered 1% aqueous uranyl acetate, and then rinsed thoroughly with distilled water. The samples were dehydrated with increasing concentrations of ethanol, then transferred to increasing concentrations of hexamethyldisilazane (HMDS) and air dried overnight. Samples were mounted on to double-stick carbon tabs (Ted Pella), which have been previously mounted onto glass microscope slides. The samples were then coated under vacuum using a Balzer MED 010 evaporator (Technotrade International) with platinum alloy for a thickness of 25 nm, then immediately flash carbon coated under vacuum. The samples were transferred to a desiccator for examination. Samples were examined/imaged in a JSM-5910 scanning electron microscope (JEOL) at an accelerating voltage of 5 kV.

## Transmission electron microscopy

Transmission electron microscopy (TEM) was performed as previously described[14,60]. Tissue samples were fixed with a solution containing 3% glutaraldehyde plus 2% paraformaldehyde in 0.1 M cacodylate buffer (pH 7.3), then washed in 0.1 M sodium cacodylate buffer and treated with 0.1% Millipore-filtered cacodylate buffered tannic acid and postfixed with 1% buffered osmium. Cultured podocytes were fixed with a solution including 0.5% glutaraldehyde plus 2% paraformaldehyde. Fixed samples were washed in 0.1 M sodium cacodylate buffer (pH 7.4) and treated with 0.1% Millipore-filtered cacodylate buffered tannic acid, postfixed with 1% OsO4/1.5% potassium ferrocyanide (KFeCN6), and stained *en bloc* with 1% Millipore-filtered uranyl acetate. The samples were polymerized in a 60 °C oven for approximately 3 days. Ultrathin sections were cut in a Leica Ultracut microtome (Leica), stained with uranyl acetate and lead citrate in a Leica EM Stainer, and examined in a JEM 1010 transmission electron microscope (JEOL) at an accelerating voltage of 80 kV. Digital images were obtained using AMT Imaging System (Advanced Microscopy Techniques).

## Mitochondrial morphological measurement

Mitochondrial morphological measurement was performed as previously described[14]. Briefly, mitochondrial aspect ratio was defined as the major and minor axes of the ellipse expressed as a fraction.

Circularity was $4\pi \times$ (mitochondrial area (Am) per [perimeter (Pm)]$^2$), and roundness was $(4 \times Am)/(\pi \times [\text{major axis}]^2)$. Feret was the longest distance between any two points along mitochondrial perimeter.

## Cristae morphological assessment

Mitochondrial cristae morphology was assessed using TEM micrographs and analyzed with ImageJ software. To quantify mitochondrial cristae abundance, the outer mitochondrial membrane, the inner mitochondrial membrane, and the individual crista were traced using the Freehand tool in ImageJ. Total cristae length and mitochondrial area were defined as the sum of the length of individual crista and the area surrounded by outer mitochondrial membrane, respectively. To determine the cristae and cristae junction number, the numbers of cristae and cristae junction in each mitochondrion were counted[61,62].

## Immunofluorescence staining

Podocytes seeded on cover slips were washed with cold PBS, fixed in 4% formaldehyde, and permeabilized with 0.1% Triton X-100 (Acros). Cells were blocked in a solution containing 50 mM Tris pH 7.6, 1% BSA (Jackson Immuno-research), and 155 mM sodium chloride (TBS) with 0.1% Triton-X-100. The cells were incubated overnight at 4 °C with appropriate primary antibodies in blocking buffer, then washed 3 times in TBS and incubated with appropriate secondary antibodies in the blocking buffer for 1–2 hrs at RT. Cells were washed 3 times in TBS and mounted onto slides. Images were captured by FV1200 MPE confocal microscope (Olympus). Quantification was carried out using Image J. Antibodies and dyes used in this experiment are summarized in Supplementary Table 3.

## Comparative mitochondrial proteome and complexsome profiling

Mitochondria enriched fractions isolated from podocytes were digested in a buffer containing 50 mM ammonium bicarbonate with LysC enzyme for 2 h at room temperature followed by Trypsin enzyme digestion at 37 °C overnight. The digestion was neutralized by 0.5% final formic acid (FA) and the peptides were measured using the Pierce™ Quantitative Colorimetric Peptide Assay. The peptides were subjected to simple C18 clean up, and LC-MS/MS analysis was carried out using a nano-LC 1200 system coupled to Orbitrap Lumos ETD mass spectrometer (Thermo Fisher). 1 µg peptide was loaded on a two-column setup with precolumn (2 cm × 100 µmI.D.) and analytical column (20 cm × 75 µmI.D.) filled with Reprosil-Pur Basic C18 (1.9 µm, Dr. Maisch GmbH, Germany) as described previously[63]. The MS raw data was searched using Proteome Discoverer 2.1 software (Thermo Fisher) with Mascot algorithm against mouse NCBI refseq database updated 2020_0324. The precursor ion tolerance and product ion tolerance were set to 20 ppm and 0.5 Da, respectively. Maximum cleavage of two with Trypsin enzyme, dynamic modification of oxidation (M), protein N-term acetylation and deamidation (N/Q) were allowed. For statistical assessment, missing value imputation was employed through sampling a normal distribution N (µ-1.8 σ, 0.8σ), where µ, σ are the mean and standard deviation of the quantified values. The median normalized and $\log_{10}$ transformed iBAQ values (Adj-iBAQ) were used for data analysis. Targets with low Adj-iBAQ values (<6.0) in the mitochondrial protein from WT and *Ins2*[Akita/+] mice were excluded from the analysis, because these low abundant proteins greatly exaggerated the ratios between *Ins2*[Akita/+] and WT. For complexsome profiling, digitonin-solubilized mitochondria proteins were subjected to BN-PAGE analysis. Five putative SC bands, determined as described previously[39] and in Supplementary Fig. 4f, were excised and cut into 1×1 mm pieces followed by in-gel digestion using LysC and trypsin enzymes. The peptides were dried in a speed vac and dissolved in 10 µl of 5% methanol containing 0.1% FA buffer. LC-MS/MS analysis was conducted in the same way as described above. The peptides identified from mascot result file were validated with 5% false discover rate (FDR).

The gene product inference and quantification were done with label-free iBAQ approach using 'gpGrouper' algorithm[64].

## APEX proximity labeling

APEX proximity labeling was performed as previously described[65,66]. In brief, 80–90% confluent cells were pretreated with 500 μM biotin-tyramide for 30 min, followed by 1 mM $H_2O_2$ activation for 1 min. After quenching and washing with PBS containing 5 mM Trolox and 10 mM sodium ascorbate, cells were harvested into PBS. The cells were centrifuged, and cell pellets were lysed in RIPA buffer with 1 mM PMSF, 5 mM Trolox, 10 mM sodium ascorbate, and 10 mM sodium azide. Cell lysates were incubated with prewashed Pierce streptavidin magnetic beads (Thermo Fisher) overnight in a cold room. The beads were washed with RIPA buffer (Teknova) twice, with 1 M KCl, 0.1 M $Na_2CO_3$, and 2 M urea in 10 mM Tris-HCl (pH 8.0) once, and with RIPA buffer twice. Bound biotinylated proteins were eluted in 1 × SDS sample buffer and separated on 4–20% PAGE, followed by immunoblotting with NDUFS4 antibody. The immuno-precipitated samples were resolved on NuPAGE 10% Bis-Tris Gel. Each lane was excised into 4 equal pieces and combined into two tubes after in-gel digestion using LysC and trypsin enzymes. Digested peptide was processed and used for the LC-MS/MS analysis in the same way as complexsome profiling. For differential analysis, we used the moderated t-test and $\log_2$ fold changes as implemented in the R package limma and multiple-hypothesis testing correction was performed with the Benjamini–Hochberg procedure.

## Co-immunoprecipitation (Co-IP)

HEK293T cells were transiently transfected (Lipofectamine 2000, Invitrogen) with constructs expressing FLAG-NDUFS4 and HA-STOML2 (both tagged at C-terminus) and crosslinked with 1 mM dithiobis (succinimidyl propionate) (Sigma) for 30 min and quenched with 50 mM Tris (pH 8.0) for 3 min[67]. Cell pellets from scraped suspension were lysed in 0.2 ml RIPA buffer (Teknova) on ice for 15 min, diluted with 0.8 ml of NETN buffer (170 mM NaCl, 1 mM EDTA, 50 mM Tris, pH 7.3, 0.5% NP-40) and incubated for another 15 min. Cleared lysate were then incubated with Anti-FLAG M2 Magnetic Beads (Sigma) overnight at 4 °C. Bound complexes were washed 5 times with NETN buffer and eluted with 100 μg/ml of 3X FLAG peptide (Sigma), separated on SDS-PAGE and analyzed by immunoblotting.

## GST pull-down assay

HEK293T cells were transfected (Lipofectamine 2000, Invitrogen) with constructs expressing C-terminus HA-tagged STOML2 wild type and deletion mutants and lysed in TNTE buffer (10 mM Tris HCl, pH 7.8, 150 mM NaCl, 1 mM EDTA, and 1.0% Nonidet P-40)[50]. Clear lysate was incubated at 4 °C for 3 h with 2 μg GST or GST-NDUFS4 protein immobilized on Glutathione Sepharose 4B beads (GE Healthcare). GST-NDUFS4-bound complex was washed in TNTE buffer five times, separated on SDS-PAGE and analyzed by immunoblotting.

## STED/STORM imaging

STED imaging was performed on an STEDYCON (Abberior) and Eclipse Ti2 inverted imaging system (Nikon Instruments) with a Plan Apochromat 100X (NA 1.49) oil immersion objective (Nikon Instruments). The STEDYCON STED unit was equipped a with 561 nm and 640 nm pulsed excitation lasers and a 775 nm pulsed depletion laser and adjustable pinhole (64 μm), and fluorescence detected by APD detectors after spectral filtering for the orange (650–710 nm) and red channels (550–615 nm), respectively. For imaging of NDUFS4, STAR-Orange was excited at 561 nm and its fluorescence emission was detected at 616 nm with 1–7 ns time gating and STAR-RED excitation used for STOML2 at 640 nm and the emission collected at 640 nm with the same time gating. The two channels were acquired sequentially, using pixel dwell times of 17 μs with a 15 nm pixel size. Mander's coefficients[68] were calculated as the indexes of intensity based

colocalization, and distance of particle's center between NDUFS4 and STOML2 were obtained using JACoP plugin of ImageJ[69]. Object-based colocalization was defined as a distance less than 140 nm[70]. For STORM imaging, imaging experiments were conducted with a N-STORM (Nikon Instruments) on an Eclipse Ti2 inverted microscope. STORM images were collected in a 512 ×512-pixel region of interest using a CFI Apochromat TIRF 100x (NA 1.49) oil object (Nikon Instruments) and a C11440-22CU ORCA-flash sCMOS 4.0 V2 camera (Hamamatsu). Images were acquired sequentially 500 frames per filter channel at 200 ms time duration. Cells labeled with Alexa Fluor 647 and Atto 488 secondary antibodies were excited with 100% laser power from a 647 nm and a 488 nm laser, respectively. Nikon Nd2 files were separated and converted to tiff files per channel by custom python script. Single-molecule localization and nearest neighboring distance (NND) were determined by using ThunderSTORM plugin of image J[71] with default setting except the camera setting: pixel size is 65 nm, photoelectrons per A/D count is 0.57. A set of post processing methods with default setting were applied to optimize the localization data. In each cell, median NND and % colocalization were calculated using a total of data in four non-overlapping representative regions of 20 × 20 μm². Maximum intensity projection (MIP) images were reconstructed from Nikon Nd2 files using Huygens Essential (Scientific Volume Imaging).

## Cryo-ET

Cryo-grids were prepared by plunge-freezing in a liquid ethane using a Vitrobot Mark IV (Thermo Fisher) that was set to 100% humidity at 4 °C. 2 μl of purified mitochondria sample was applied to 200 mesh, R 2/1 Quantifoil copper grids (Quatifoil) and blotted with Whatman filter paper for 4 s. Cryo-ET data collection was performed using a Titan Krios G3 300 keV FEG transmission electron cryo-microscope (Thermo Fisher). Cryo-ET images were acquired using a BioQuantum energy filter (Gatan) with slit width set to 20 eV. Images were recorded on a 4k × 4k K2 Summit direct electron detector (Gatan) operated in counting mode at nominal microscope magnifications of 26,000×, 33,000× or 19,500× corresponding to pixel sizes of 5.32 Å, 4.20 Å and 7.50 Å for NG, HG and HG-DOX, respectively. SerialEM software[72] was used for imaging. Each tilt series was collected from −50° to +50° with increment of 2° using the low dose functions for tracking and focusing. The cumulative dose of each tilt-series was 80–90 e⁻/Å². Defocus values were set between −8 μm and −10 μm. For tomogram reconstruction and segmentation, raw movie frames of tilt-series were corrected for beam-induced motion using MotionCor2[73]. The aligned micrographs were imported into EMAN2[74] and were compiled into tilt-series. Automated alignment was performed and 1k × 1k 3D tomograms (bin4) were generated using e2tomogram.py in EMAN2 software package[74]. The references containing features of interest were manually boxed out in tile images of 64 ×64 pixels using EMAN2 graphical tool and then used for training Convolutional neural network (CNN). Once a CNN has been trained to recognize a certain feature, it was applied to annotate the same feature in a tomogram.

## Protein structure prediction and molecular docking

Human STOML2 structure was predicted by Contact-guided Iterative Threading ASSEmbly Refinement web tool (C-I-TASSER)[75]. Human NDUFS4 within the structure of the respirasome was obtained from the PDB database (5XTB)[76]. The ClusPro 2.0 web server[44] was used to perform protein-protein docking simulation. The resulting docking structures were analyzed using PyMol Molecular Graphics Systems, version 2.0, Schrodinger, LLC. CryoEM resolved RSC structure ($I_1III_2IV_1$, PDB-5XTH) was visualized using the UCSF ChimeraX[43].

## Statistics

Group data are expressed as mean ± SEM or median ± IQR. Comparisons between two groups were performed using two-tailed Student's $t$ test for normally distributed data and two-tailed Mann–Whitney test

for non-normally distributed data. Categorical variables between two groups in human data were compared with the Fisher exact test. Comparisons of multiple groups were performed using one way-analysis of variance (one-way ANOVA) followed by Tukey–Kramer's multiple comparisons test for normally distributed datasets and Kruskal–Wallis followed by Dunn's multiple comparisons test for non-normally distributed datasets. Multiple paired *t*-test were followed by Holm-Sidak test, and multiple unpaired Student's *t* test were followed by two-stage linear step-up procedure with the FDR Q = 0.05. All tests were performed with GraphPad version 9.3.1 (Graphpad Software), and *P* values < 0.05 were considered statistically significant.

## Reporting summary

Further information on research design is available in the Nature Portfolio Reporting Summary linked to this article.

## Data availability

All data supporting the findings of this study are available in the main text or supplementary materials. The mass spectrometry proteomic data for mitochondrial proteome in murine podocytes, NDUFS4-APEX proteome, and murine podocyte complexsome have been deposited to the ProteomeXchange Consortium via the PRIDE partner repository with the dataset identifiers, PXD041202, PXD045828, PXD041378, and PXD041203. Human NDUFS4 within the structure of the respirasome and the structure of the I$_1$III$_2$IV$_1$ within RSC were obtained from the PDB database 5XTB and 5XTH, respectively. Glomerular transcriptomic data (Nephroseq database version 5) were analyzed using Ju CKD Glom dataset (GSE104948). Source data are provided with this paper.

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

## Acknowledgements

We thank Drs. Anna Malovannaya and Antrix Jain at the Mass Spectrometry Proteomics Core Facility of Baylor College of Medicine (BCM) for their assistance in proteomic and complexsome profiling experiments (NIH P30CA125123 and CPRIT Core Facility Award RP210227), High Resolution Electron Microscopy Facility at the University of Texas MD Anderson Cancer Center (UT-MDACC) for TEM and SEM studies (CCSG NIH P30CA016672), Advanced Microscopy Core at UT-MDACC for laser scanning confocal microscopy and imaging analysis (CPRIT Award

RP170628), Metabolomics Core at UT-MDACC for the Seahorse analysis (CPRIT Award RP130397 and NIH grants S10OD012304-01 and P30CA016672), BCM Genetically Engineered Rodent Models Core for generating the Ndufs4 transgenic mouse, the Center for Advanced Microscopy, a Nikon Center of Excellence, in the Department of Integrative Biology & Pharmacology at the University of Texas Health Science Center at Houston for the technical assistance with STED and STORM image analyses (NIH/NHLBI HL143111), the Cryo-EM Core Facility at the UTHealth at Houston (NIH R01GM072804 (to I.I.S.) and CPRIT RP190602 (to I.I.S. & S.J. Ludtke/Baylor College of Medicine) Awards) for the aid with Cryo-ET experiments, and the Flow Cytometry & Cellular Imaging Facility at UT-MDACC (NIH P30CA016672). We also thank Prof. Kenichi Ohashi at the Department of Human Pathology at the Tokyo Medical & Dental University for evaluating renal pathology in human DKD and donor kidneys. This study was supported by the National Institute of Diabetes and Digestive and Kidney Diseases grants R01DK078900 and R01DK091310 (to F.R.D). K.M. was a recipient of awards from the Manpei Suzuki Diabetes Foundation, Wesco Scientific Foundation, the Ichiro Kanehara Foundation for the Promotion of Medical Sciences and Medical Care, the Uehara Memorial Foundation, and the Japan Society for the Promotion of Science.

## Author contributions

F.R.D. and B.H.C designed this study and wrote the original draft. K.M. and D.L.G. performed the animal experiments, managed the mouse colony, and collected samples. D.L.G. and J.L. created the Ndufs4 construct; K.M., M.A., and J.W. collected human samples and performed histological analysis; K.M., J.L., D.L.G., and Z.Y., carried out proteomic, BN-PAGE, and complexome analyses. K.M., C.R.J, and M.A.Z. conducted mito-roGFP imaging analyses. K.M., R.S., G.F., and I.I.S performed CryoET. K.M. and T.I.M. conducted super-resolution imaging analyses. K.M., J.L., and D.L.G. conducted APEX analysis. K.M., J.L., D.L.G., and B.H.C completed additional wet lab experiments. F.R.D. and B.H.C supervised all experiments and data analyses. F.R.D., P.T.S., B.H.C., K.M., and J.L., wrote the manuscript with consultation from all authors.

## Competing interests

The authors declare no competing interests.
