## [Peer Review File · Nature Communications]

NDUFS4 Regulates Cristae Remodeling in Diabetic Kidney DiseaseREVIEWER COMMENTS

Reviewer #1 (Remarks to the Author):

This paper investigates the interesting hypothesis that the NDUFS4 accessory subunit of OXPHOS complex I is involved in the control/remodeling of mitochondrial cristae architecture. Conceptually, it presents extensive experimental data focusing on effects in NDUFS4 overexpression systems, as well as the potential interaction between NDUFS4 and STOML2. Although the scientific message of this paper certainly can be appealing, I still have several questions/concerns related to the used experimental strategies and proposed concepts.

Fig. 1b: it is unclear to me which CI subunits were significantly reduced

Fig. 1c: what does the Z-score indicate?

Fig. 1d (and others): why is CI activity expressed as NAD⁺/NADH? This ratio is dependent on many other metabolic pathways. For proper analysis of (rotenone-sensitive) CI activity more "classical" enzymatic assays in mitoplasts should be used.

Fig. 1h: Why was only NDUFS4 analysed? I guess there should also be significant correlations with other CI subunits? When such correlations are absent, this would strengthen the choice to focus on NDUFS4 for further analysis. When such correlations are present, why was NDUFS4 selected?

Fig. 1i-l: more details are needed on how NDUFS4 staining was exactly quantified

Page 6, lines 8-10: in my view, the data suggests that CI expression is important, meaning that there is no evidence for NDUFS4-specific effects.

Fig. 3a: please compute the relevant respiratory control ratios (RCRs). These are independent of cell number and will highlight functional defects. Furthermore, there is no ECAR data presented. Why?

Fig. 3g (bargraphs): the average WT value is 1.0; how was data normalization performed for this and all relevant other figures?

Fig. 3h-n (and page 7; line 23): the presented parameters do not reflect mitochondrial "dynamics" but steady-state mitochondrial morphology. One can not talk about enhanced fission (it can also be reduced fusion) since mitochondrial fission/fusion proteins were not investigated. Moreover, it is unclear how the conclusion that "cristae morphology is altered" is reached.

Fig. 3m (page 8; line 2): MitoSOX does not specifically detect "mitochondrial ROS". The ethidium molecule (which is the ROS-detection moiety of MitoSOX) reacts extremely fast (but not exclusively with) superoxide. This means that MitoSOX can become fluorescent in the cytosol during its transit to the mitochondrial matrix (driven by the presence of the decylTPP moiety of MitoSOX).

Fig. 3n (page 8; line 2): the used assay reports on total cellular ATP content, so is not informative on ATP production.

page 8; line 6: what is meant with "mitochondrial reprogramming"?

Page 8; line 7: please rule out that the used DOX concentration and incubation regime impacts on mitochondrial function (described in the literature).

Page 8; line 10: it is not explained why HG is compared to NG and why this is relevant for the rest

of the study.

Fig. 4f: what are all the other (non-marked) bands representing. Or is this not relevant? Please explain. I guess the AB cocktail was also used here?

Page 9; line 7: why is it surprising that NDUFS4 OE restored CI in-gel activity "even under HG conditions"?

Page 9; line 11: that NDUFS4 exerts a "regulatory effect" is not demonstrated by the preceding experimental results.

Regarding the STOML2-related experiments, I wonder if the proposed binding of STOML2 to NDUFS4 protein is compatible with the currently available structural information on the CI holocomplex and CI in a respiratory supercomplex? In my opinion, this is the most potentially interesting finding of this work since all functional effects of NDUFS4 OE could be explained by the NDUFS4 increase leading to more fully assembled CI. How does the cell "know" that more other CI subunits are required in the NDUFS4-overexpressing case? This would mean that NDUFS4 might be (among) the rate-limiting subunits for CI assembly. Is this supported by the current literature on CI biogenesis?

Reviewer #2 (Remarks to the Author):

The paper by Mise et al has explored the role of the Complex I subunit Ndufs4, in the pathogenesis of diabetic kidney disease. The aim of this study was to assess ETC abundance, mitochondrial function and morphology in kidney podocytes in a mouse model of diabetes and whether this affected DKD progression. First, ETC abundance was determined by mass spectrometry in two mouse models of diabetes, the Ins2Akita mouse and the db/db mouse model. Complex I abundance and mRNA expression was decreased in podocytes of the Ins2Akita mouse and db/db mouse.

Out of the several downregulated subunits of mitochondrial OXPHOS complexes, Ndufs4 was chosen as a target. Glomerular Ndufs4 protein was decreased in human biopsies from patients with DKD. Next, podocyte-specific Ndufs4-transgenic mice were generated and crossed onto the Ins2Akita model of type 1 diabetes. Podocyte-specific Ndufs4-transgenic mice were partially protected from the diabetic kidney disease phenotype. Similar data were shown in the db/db mouse model. Ndufs4 overexpression in podocytes rescued high glucose-induced mitochondrial cristae changes as well as supercomplex assembly. A molecular interaction was identified between STOML2 and NDUFS4. The authors conclude that the findings represent a major paradigm shift in the current management of DKD by suggesting that targeting ETC remodeling could be a promising approach for developing therapies to mitigate the progression of DKD.

Major comments

ETC remodeling, mitochondrial function and morphology in kidney podocytes in rodent models with diabetes has been widely reported, including from the current group. Several other studies have explored the role of mitochondrial Complex I subunits, leading to ETC remodeling in the development of chronic kidney disease (e.g., PMID: 23320803), therefore the idea itself is not novel. These papers should be referenced. Many others have suggested that targeting the electron transport chain is an approach to treat CKD/DKD (PMID: 36781216), therefore the author's suggestion of a major paradigm shift should be moderated.

It is not clear which method was used to determine Complex I activity. The method referenced in the paper (ref 43), reports using NBT on tissue sections. This is not a generally accepted method. The gold standard method used by mitochondrial laboratories for determination of complex I activity is via respiratory chain enzymology using kinetic spectrophotometric assays (Frazier AE et al., Assessment of mitochondrial respiratory chain enzymes in cells and tissues, *Methods Cell Biol*

2020). Each assay requires specific inhibitors to ensure that the assay reflects true activity of the specific respiratory chain enzymology complex being assayed, i.e., rotenone-sensitive CI activity. In order to avoid misinterpretation, the mitochondrial marker enzyme citrate synthase is typically assayed to enable correction for mitochondrial content by expressing enzymes as citrate synthase ratios. Complex I activity should be determined using the kinetic spectrophotometric assay outlined in Frazier.

"We also evaluated NDUFS4 staining in glomeruli from diabetic subjects with a wide spectrum of DKD histology, and found that NDUFS4 staining in glomeruli was progressively reduced with worsening of DKD histology (test for trend $P < 0.01$) (Fig. 1l)." Was there a significant difference from Class I CKD to Class III CKD, or between any of the DKD classes? Can you explain the Test for trend? Is it really a progressive decrease in glomerular Ndufs4 staining?

Fig1K. Statistical analysis was performed on a sample size of $n=2$ within the microalbuminuria group. How is this possible when using one way-analysis of variance followed by Tukey's multiple comparisons test in Graphpad Prism? Please confirm the statistical test used.

What was the mitochondrial density of the podocytes in diabetic mice? Ins2Akita and db/db mice.

Fig 3n, ATP production. What are the units of the ATP assay? Both the mitosox and ATP data are expressed as relative to control. As is the Complex I. The ATP assay used appears to be a one-step single reagent assay for detection of viable cells in culture, rather than an assay which can reliably determine ATP production over time in a sensitive manner. I would caution against using such an assay when screening for differences between groups. Was a standard curve generated and used? Did the authors account for cell number or total protein? The same questions apply to the mitosox assay and data. Representative histograms showing mean intensity of MitoSOX fluorescence should be shown for each group.

Minor comments

Some of the individual data points on graphs are missing through the manuscript (e.g., Fig 2h, k, l & m; Figure 3g, i, j, k, l) and throughout the paper.

Fig 2a. What sample is contained in each lane in the genotyping gel?

Fig 2b. The mRNA expression of Ndufs4 appears higher in the "podocyte depleted" fraction of renal cortex vs the podocyte fraction. It is important to show Ndufs4 protein in podocytes vs tubules and total renal cortex.

For the Seahorse cell culture studies, how many independent cell culture experiments are shown? It is stated that there were $n=6-8$ replicates per group. Are these data derived from one cell culture experiment?

Extended data figure 5a. The western blots are not convincing. The quantitation needs to be shown. In mouse kidney, OPA1 generally has 5 isoforms. There should be 5 bands present. Samples need to be run on a gradient gel to reveal the five isoforms as per PMID: 26822084.

Introduction. Page 4, line 13. "ETC dysfunction is recognized as an important cause of organ failure in several human pathologies including heart failure, diabetes, and neurodegeneration in a tissue-specific manner. Mitochondrial cytopathies have long been recognised to lead to kidney failure. This should be mentioned in the introduction (PMID: 33305107)

Reviewer #3 (Remarks to the Author):

In this manuscript, Mise et al. examine the importance of the mitochondrial ETC Complex I protein Ndufs4, in the maintenance of mitochondrial structure in the context of the diabetic kidney. The Authors show that Ndufs4 plays a role in the regulation of cristae structure and supercomplex formation, while providing evidence of binding partners in the process. The studies address an interesting topic, and the experimentation appears thorough and well-executed. Specific comments are indicated below.

1) It is not entirely clear why the Authors chose to focus on Ndufs4. The heat map in Fig.1 shows many ETC proteins, including in Complex I that were also decreased as a result of diabetes. In fact, there are some that were decreased by a greater extent. The decision to choose Ndufs4 comes across as a biased decision without scientific justification. A greater case should be made for choosing this target. In the Discussion, the Authors indicate that examination of other subunits would be meaningful, and this is appreciated, but does not preclude the need for a greater justification for the Ndufs4 focus.

2) Introduction, page 4; in the last sentence, the Authors state that their "results unexpectedly reveal that ETC integrity determines the stability of RSCs." This seems disingenuous. Do the Authors really believe that the very integrity of the components of the ETC would not be an important factor in the stability of the supercomplex in which they reside? My suggestion would be to either remove or soften this statement.

3) The use of multiple experimental models of diabetes (Akita, db/db) and human patient samples is appreciated and strengthen the conclusions and relevance of the studies. With that being said, the inclusion of complete db/db data (especially proteomic profiling in Fig.1b) is glaringly absent. Inclusion of this data makes a stronger case for commonality in the models and the ultimate focus on Ndufs4.

We greatly appreciate the comments/suggestions made by the reviewers. In the revised version of our manuscript, we have made a number of additional experiments that are now presented in 15 new panels that include:

- Rotenone-sensitive complex I enzymatic activity (**Figs. 1d,2d; and Extended Data Fig. 4g**)
- Real-time ATP production (**Figs. 3p,3q**),
- Mitochondrial ROS assessment using mito-roGFP, a specific mitochondrial matrix-targeted redox-sensitive reporter (**Fig. 3o**)
- Proteomic data from type 2 diabetic mice *Lepr^{db/+}* and *Lepr^{db/db}* mice (**Figs. 1e and Extended Data Fig. 1d**),
- RCR and ECAR data (**Extended Data Figs. 3a,3b**)
- GST-NDUFS4 pulldown assay to identify more specific binding sites between NDUFS4 and STOML2 (**Figs. 6g,6i**).
- Excluding the DOX effect in our DOX-inducible model (**Extended Data Figs. 4c, 4d, 4e**)

The additions to the manuscript are highlighted in yellow for ease of the reviewers.

Reviewer #1

Comment 1: Fig. 1b: it is unclear to me which CI subunits were significantly reduced. In response to the reviewer's comment, we have since introduced an informative table (Extended Data Fig 1d) in addition to the previously shown Fig 1b. This table serves to outline the CI subunits with significantly reduced values, providing a convenient and comprehensive reference for those specific CI subunits.

Comment 2: Fig. 1c: what does the Z-score indicate?

In the original manuscript, we employed Z-scores, a widely used statistical tool for proteomic analysis, primarily to indicate the extent to which a specific data point is deviated from the mean, measured in terms of standard deviations. In the revised version of the manuscript, we adopted an alternative approach by using adjusted intensity-based absolute quantification (iBAQ) values. Adjusted iBAQ values represent protein abundance estimates that have been normalized for meaningful comparison between samples^{1,2} (Cabrera-Orefice A. et al Front Cell Dev Biol 2022; 9:796128 and Välikangas T. et al., Brief Bioinform 2018;19(1):1-11). Both methods are commonly used in proteomic analysis, but they provide different types of information. Whereas Z-scores can help identify proteins of interest based on their deviation from the mean, adjusted iBAQ values provide information about the relative abundance of proteins across different conditions or samples. Furthermore, we have now included heatmaps generated using these adjusted iBAQ values for each complex subunit (Extended Data Figure 1b).

Comment 3: Fig. 1d (and others): why is CI activity expressed as NAD⁺/NADH? This ratio is dependent on many other metabolic pathways. For proper analysis of (rotenone-sensitive) CI activity, more "classical" enzymatic assays in mitoplasts should be used.

We appreciate the reviewer's comment. In the revised version of the manuscript, we have assessed CI activity using the classical kinetic spectrophotometric assays³ (Frazier AE et al. Methods Cell Biol 2020;155:1221-156) (Fig. 1d and Fig. 2d.). Our findings closely mirror the

results previously presented in our initial submission, as illustrated in Extended Data Fig. 1c, Extended Data Fig. 2a, and Extended Data Fig. 4d. The initial assessment of CI activity was based on a well-established commercially available assay (Abcam, ab109721) used in other publications⁴⁻⁶ (Wang T. et al., *Cell Metab* 2021; 33 (3):531-546.e9; Balsa E., et al., *Mol Cell* 2019; 74 (5):877-890.e6; and Cao LL. et al., *Nature* 2016; 539(7630):575-578).

Comment 4: Fig. 1h: Why was only NDUFS4 analyzed? I guess there should also be significant correlations with other CI subunits? When such correlations are absent, this would strengthen the choice to focus on NDUFS4 for further analysis. When such correlations are present, why was NDUFS4 selected?

We selected NDUFS4 among several CI subunits in our proteomic experiment for several compelling reasons: 1. Proteomic Analysis: in our comprehensive proteomic investigation, we identified NDUFS4 as a significantly downregulated subunit within CI in kidney podocytes from diabetic Akita mice, in comparison to those from a control group of non-diabetic mice. 2. Consistent Validation: subsequent validation studies, spanning various models including cellular, murine, and human samples, consistently revealed NDUFS4 as the most prominently downregulated subunit in diabetes when compared to nondiabetic controls, further substantiating its significance. In contrast, several other potential CI candidates, including NDUFA2, NDUFB3, NDUFB4, NDUFB5, NDUFB8, NDUFB11, and NDUFV3 were not consistently downregulated in the diabetic mice or in the glomeruli of patients with DKD (Extended Data Fig. 1e,f,g). 3. Early Downregulation: examination of human samples of individuals with DKD indicated that the downregulation of NDUFS4 occurs prior to the onset of albuminuria in patients, suggesting its potential role as an early marker of dysfunction (Fig. 1k,i). 4. Genetic Evidence: importantly, both human and murine genetic data underscore the critical role of NDUFS4 in mitochondrial function since the absence of functional NDUFS4 is associated with Leigh syndrome, emphasizing its indispensable function within mitochondrial processes.

Comment 5: Fig. 1i-l: more details are needed on how NDUFS4 staining was exactly quantified.

In the revised version of our manuscript, we have incorporated comprehensive information regarding the staining and quantification of NDUFS4 ("Methods" section on page 33). To quantify NDUFS4, we computed the mean intensity within the glomerular tuft area across all glomeruli obtained from healthy donor and DKD kidney samples. This quantification was carried out utilizing Image J software, following an established protocol previously outlined⁷ (Falkevall A, et al. *Cell Metab* 2017;25(3):713-726).

Comment 6: Page 6, lines 8-10: in my view, the data suggests that CI expression is important, meaning that there is no evidence for NDUFS4-specific effects.

Our initial data indicate a significant downregulation of CI subunits within the podocytes of diabetic mice, as demonstrated in Fig. 1c. However, among these CI subunits, Ndufs4 consistently displayed reduced expression levels in both diabetic mouse models and human patients. It is important to clarify that our study at that point simply highlighted the potential significance of Ndufs4 in the context of our experimental approach.

Comment 7: Fig. 3a: please compute the relevant respiratory control ratios (RCRs). These are independent of cell number and will highlight functional defects. Furthermore, there is no ECAR data presented. Why?

We appreciate this concern raised by the reviewer. We have added RCRs values to the revised version of the manuscript. Consistent with our previous data, we observe that RCR is

significantly reduced in primary podocytes from diabetic *Ins2^{Akita/+}* mice compared to those in WT mice and *Ins2^{Akita/+}-Ndufs4^{podTg}* mice. We have added these data to the Extended Data Fig. 3a. Regarding ECAR, we have also added ECAR data into Extended Data Fig. 3b.

Comment 8: Fig. 3g (bargraphs): the average WT value is 1.0; how was data normalization performed for this and all relevant other figures?

We quantified the NADH oxidase staining using Image J software. For each glomerulus (a total of 60 glomeruli per group), we determined the intensity divided by the median intensity observed in the WT (wild type) images. This standardized approach ensures that the median intensity of the WT group is set to 1.0. The same normalization process was applied in Extended Data Fig. 2a to assess the relative CI activity in glomeruli.

Comment 9: Fig. 3h-n (and page 7; line 23): the presented parameters do not reflect mitochondrial "dynamics" but steady-state mitochondrial morphology. One cannot talk about enhanced fission (it can also be reduced fusion) since mitochondrial fission/fusion proteins were not investigated. Moreover, it is unclear how the conclusion that "cristae morphology is altered" is reached.

We appreciate this insightful question raised by the reviewer and agree that mitochondrial dynamics predominately depend on fission, fusion, shape transition, and transport or tethering along the cytoskeleton. Consistent with these concepts, we have previously shown that kidney podocytes exposed to high glucose conditions exhibit a fragmented or punctate phenotype^{8,9} (Wang W. et al., *Cell Metab* 2012;15(2):186-200; and Galvan DL., et al., *J Clin Invest* 2019;129(7):2807-2823). Our earlier work also has clearly demonstrated that this shift towards shorter and fragmented mitochondria in high glucose-treated podocytes is attributed in part to an enhanced mitochondrial fission process characterized by increased Drp1 activity, the primary protein governing mitochondrial fission⁸ (Wang W. et al., *Cell Metab* 2012;15(2):186-200). In a subsequent study, we developed mouse models where the phosphorylation site of DRP1 at Ser 600 was mutated and rendered inactive specifically in podocytes of diabetic mice⁹ (Galvan DL., et al., *J Clin Invest* 2019;129(7):2807-2823). Additionally, we reported an increase in actin/Drp1 interactions associated with DRP1 phosphorylation⁹ (Galvan DL., et al., *J Clin Invest* 2019, 129(7):2807-2823). Therefore, the mitochondrial morphology depicted in Fig. 3h, particularly the observed shortening of mitochondria in the kidney podocytes of diabetic *Ins2^{Akita/+}* mice, is consistent with our prior publications in various experimental models of diabetic kidney disease. We concur with the reviewer that mitochondria maintain their morphology through a dynamic balance of fission and fusion processes in cells which is regulated by a number of regulatory kinetic proteins. Regarding cristae morphology, we quantified mitochondrial cristae abundance in transmission electron microscopy (TEM) micrographs obtained from primary podocytes using Image J. Our analysis revealed a significant reduction in cristae abundance in podocytes from *Ins2^{Akita/+}* mice, which was subsequently improved in the *Ins2^{Akita/+}-Ndufs4^{podTg}* mice. These results have been included in Fig. 3m and n.

Comment 10: Fig. 3m (page 8; line 2): MitoSOX does not specifically detect "mitochondrial ROS". The ethidium molecule (which is the ROS-detection moiety of MitoSOX) reacts extremely fast (but not exclusively with) superoxide. This means that MitoSOX can become fluorescent in the cytosol during its transit to the mitochondrial matrix (driven by the presence of the decylTPP moiety of MitoSOX).

We appreciate and acknowledge that the use of MitoSOX, a widely employed tool for assessing mitochondrial ROS (mROS) in numerous studies^{10,11} (Sutandy F.X.R., et al., *Nature* 2023;618(7966):849-854, Labuschagne CF., et al., *Cell Metab* 2019;30(4):720-

734.e5), may lack specificity in detecting mROS. In collaboration with Dr. Paul Schumacker, a co-author of this manuscript and an internationally known expert in the area of ROS¹² (Murphy et al. **Guidelines for measuring reactive oxygen species and oxidative damage in cells and in vivo**. Nat Metab. 2022;4(6):651-662), we adopted a more precise approach for evaluating mROS in the revised version of our manuscript. We utilized a ratiometric mitochondrial matrix-targeted redox-sensitive reporter, mito-roGFP, which offers several advantages over traditional methods of assessing mROS, as previously documented by our group and others^{13,14} (Galvan DL. et al., Kidney Int 2017;92(5):1282-128; Waypa GB. et al., Circ Res 2010;106(3):526-535). The mito-roGFP reporter is highly sensitive to changes in the redox state within the mitochondrial matrix. To this aim, we introduced mito-roGFP into primary podocytes isolated from WT, *Ndufs4^{podTg}*, *Ins2^{Akita/+}*, and *Ins2^{Akita/+}-Ndufs4^{podTg}* mice. We quantified the ratios of oxidized mito-roGFP to reduced mito-roGFP intensities using confocal microscopy images. As illustrated in Fig 3o, we observed that the ratio of oxidized/reduced roGFP was elevated in primary podocytes from *Ins2^{Akita/+}* mice compared to WT mice while it was reduced in *Ins2^{Akita/+}-Ndufs4^{podTg}* mice.

Comment 11: Fig. 3n (page 8; line 2): the used assay reports on total cellular ATP content, so is not informative on ATP production.

We appreciate the critique by the reviewer and consequently have changed the term ATP production to ATP content (Extended Data Fig. 3n and Extended Data Fig. 4h). Moreover, we performed a real-time, label-free assay to quantify cellular and mitochondrial ATP production rates in live primary podocytes (Fig. 3p,q).

Comment 12: page 8; line 6: what is meant with "mitochondrial reprogramming"?

The concept of mitochondrial reprogramming underscores the dynamic nature of mitochondria and their ability to respond to signals that they receive from other cells or their microenvironment, resulting in changes to their function, morphology, number, and distribution within the cell. This degree of remodeling allows mitochondria to quickly adapt to their changing environmental cues.

Comment 13: Page 8; line 7: please rule out that the used DOX concentration and incubation regime impacts on mitochondrial function (described in the literature).

We appreciate the reviewer's comment. High doses of doxycycline have been previously shown to hinder protein translation of mitochondrially encoded genes—a phenomenon known as mitonuclear protein imbalance leading to defects in basal and maximal OCRs¹⁵ (Moullan N., et al., Cell Rep 2015;10(10):1681-1691). To determine the potential influence of DOX at a 200 nM concentration, as employed in our experiments, on mitochondrial function in our experimental models, we conducted an experiment comparing mitochondrial basal and maximal OCRs in podocytes treated with 200nM DOX compared to those treated with 1000 nM and 2000 nM. We found that using DOX at 200 nM, mitochondrial OCR remained largely unaffected. However, when incubated at a substantially higher dose (2000 nM), we observed significant changes in the basal and maximal OCR consistent with the findings in the literature. We have incorporated these new results into Extended Data Fig. 4c,d,e and provided a brief description in the Results section.

Comment 14: Page 8; line 10: it is not explained why HG is compared to NG and why this is relevant for the rest of the study.

High glucose (HG: 25mM glucose) is conventionally added to culture media to mimic diabetic conditions in cell models. In prior studies, both our lab and others have comprehensively evaluated the impact of various HG concentrations and treatment durations on eliciting

cellular responses in podocytes^{8,16-19} (Wang W. et al., *Cell Metab* 2012;15(2):186-200, Long J. et al., *J Clin Invest* 2016;126(11):4205-18, Qi W. et al., *Nat Med* 2017;23(6):753-762, Fu Y. et al., *Cell Metab* 2020;32(6):1052-1062.e8, and Cao A. et al., *J Clin Invest* 2021;131(10):e141279).

Comment 15: Fig. 4f: what are all the other (non-marked) bands representing. Or is this not relevant? Please explain. I guess the AB cocktail was also used here?

We appreciate this question raised by the reviewer. The AB cocktail was used as indicated on Fig. 4f. While we have presented the identities of the unmarked bands (Extended Data Fig. 4h), it is important to clarify that these bands are not pertinent to our current focus, which centers on identifying mitochondrial supercomplexes.

Comment 16: Page 9; line 7: why is it surprising that NDUF54 OE restored CI in-gel activity "even under HG conditions"?

We have removed "even" from the sentence in response to the reviewer's feedback. We were simply attempting to convey our surprise at the significant regulatory impact of Ndufs4 in modulating the effect of high glucose in several experimental experiments in this study.

Comment 17: Page 9; line 11: that NDUF54 exerts a "regulatory effect" is not demonstrated by the preceding experimental results.

Based on our comprehensive *in vivo* and *in vitro* observations, it is evident that NDUF54 exerts a regulatory influence over mitochondrial morphology, cristae remodeling, and supercomplexes formation. This assertion is supported by our findings that both the downregulation and overexpression of NDUF54 yield distinct and opposing effects on mitochondrial and cristae/mitochondria structure, effectively modulating their morphology. However, in response to the reviewer's insights, we acknowledge the need for further clarification regarding the precise regulatory role of NDUF54. Consequently, we are actively conducting additional experiments within our laboratory to delve deeper into this important aspect of NDUF54 function.

Comment 18: Regarding the STOML2-related experiments, I wonder if the proposed binding of STOML2 to NDUF54 protein is compatible with the currently available structural information on the CI holocomplex and CI in a respiratory supercomplex? In my opinion, this is the most potentially interesting finding of this work since all functional effects of NDUF54 OE could be explained by the NDUF54 increase leading to more fully assembled CI. How does the cell "know" that more other CI subunits are required in the NDUF54-overexpressing case? This would mean that NDUF54 might be (among) the rate-limiting subunits for CI assembly. Is this supported by the current literature on CI biogenesis?

We appreciate this interesting comment raised by the reviewer. Our Extended Data Fig. 4g suggests that CI is mainly incorporated into the RSC in podocytes. Therefore, we performed molecular docking analysis using the NDUF54 structural data derived from the Cryo-EM resolved respirasome structure (PDB database 5XTB; <https://www.rcsb.org/structure/5XTB>) and the predicted human STOML2 structure (Contact-guided Iterative Threading ASSEmbly Refinement web tool (C-I-TASSER; <https://zhanggroup.org/C-I-TASSER>) since the crystal structure of STOML2 has not yet been resolved. We supplemented our initial findings with STOML2 deletion mutant studies which indicate that stomatin domain of the STOML2 protein plays a central role for its binding to NDUF54. In the revised version, we provide new data to delineate further this interaction and show that within the stomatin domain, the $\beta 2$, $\beta 3$ and $\beta 4$

strands are necessary for the binding of STOML2 with NDUFS4 (Fig. 6g,i). These results are also consistent with the molecular docking model. Taken together, our findings suggest that the interaction between NDUFS4 and STOML2 is necessary for the proper maintenance of cristae morphology, RSC integrity, and ETC function in podocytes. Interestingly, while Ndufs4 overexpression alone does not appear to trigger a cellular response to recruit additional CI subunits, the context changes in high glucose conditions where our data suggest that reduced levels of NDUFS4 result in compromised interactions between Ndufs4 and STOML2. This disruption leads to a disorganized cristae platform for RSC assembly leading to decreased RSC assembly and contributing to decreased CI function and mitochondrial dysfunction. Overexpressing Ndufs4 in diabetic podocytes restores this imbalance leading to enhanced CI stability, improved mitochondrial RSCs assembly and cristae morphology. We believe that the interaction between Ndufs4 and STOML2 could be an important mechanism adapting the CI assembly and function in response to metabolic cues.

Reviewer #2

The paper by Mise et al has explored the role of the Complex I subunit Ndufs4, in the pathogenesis of diabetic kidney disease. The aim of this study was to assess ETC abundance, mitochondrial function and morphology in kidney podocytes in a mouse model of diabetes and whether this affected DKD progression. First, ETC abundance was determined by mass spectrometry in two mouse models of diabetes, the Ins2Akita mouse and the db/db mouse model. Complex I abundance and mRNA expression was decreased in podocytes of the Ins2Akita mouse and db/db mouse. Out of the several downregulated subunits of mitochondrial OXPHOS complexes, Ndufs4 was chosen as a target. Glomerular Ndufs4 protein was decreased in human biopsies from patients with DKD. Next, podocyte-specific Ndufs4-transgenic mice were generated and crossed onto the Ins2Akita model of type 1 diabetes. Podocyte-specific Ndufs4-transgenic mice were partially protected from the diabetic kidney disease phenotype. Similar data were shown in the db/db mouse model. Ndufs4 overexpression in podocytes rescued high glucose-induced mitochondrial cristae changes as well as supercomplex assembly. A molecular interaction was identified between STOML2 and NDUFS4. The authors conclude that the findings represent a major paradigm shift in the current management of DKD by suggesting that targeting ETC remodeling could be a promising approach for developing therapies to mitigate the progression of DKD.

Major comments

Comment 1: ETC remodeling, mitochondrial function and morphology in kidney podocytes in rodent models with diabetes has been widely reported, including from the current group. Several other studies have explored the role of mitochondrial Complex I subunits, leading to ETC remodeling in the development of chronic kidney disease (e.g., PMID: 23320803), therefore the idea itself is not novel. These papers should be referenced. Many others have suggested that targeting the electron transport chain is an approach to treat CKD/DKD (PMID: 36781216), therefore the author's suggestion of a major paradigm shift should be moderated

We concur with the reviewer's comment on the potential impact of mitochondrial dysfunction as a prominent factor implicated in the pathogenesis of kidney diseases, including diabetic kidney disease (DKD). However, the precise nature of mitochondrial dysfunction and the molecular mechanisms responsible for ETC dysfunction in podocytes leading to the

progression of DKD remain largely unknown. Our findings provide detailed insights into the pathobiology of mitochondrial respiration in podocytes and its central role in the pathogenesis of DKD. We provide evidence that Ndufs4 ties CI integrity to high glucose metabolic cues in the cell and the progression of DKD. While several studies have indeed suggested that compromised ETC function may serve as a risk factor for CKD, no prior investigations, to the best of our knowledge, have definitively demonstrated that improving CI structure/function can effectively reverse key features of DKD. Importantly, our study has provided strong experimental data to support an important biological function of Ndufs4 in preserving the integrity of cristae morphology. This novel function, in turn, exerts a protective influence on the progression of DKD. While previous studies have explored the regulatory impact of cristae-shaping proteins on supercomplexes, our work goes a step further by demonstrating for the first time that the overexpression of Ndufs4 in the diabetic milieu yields modulatory effects on cristae-shaping proteins and cristae morphology. We firmly believe that our study has introduced a completely new area of research, shedding light on the interplay between mitochondrial complex subunits and cristae forming proteins in the context of kidney pathology.

Comment 2: It is not clear which method was used to determine Complex I activity. The method referenced in the paper (ref 43), reports using NBT on tissue sections. This is not a generally accepted method. The gold standard method used by mitochondrial laboratories for determination of complex I activity is via respiratory chain enzymology using kinetic spectrophotometric assays (Frazier AE et al., Assessment of mitochondrial respiratory chain enzymes in cells and tissues, Methods Cell Biol 2020). Each assay requires specific inhibitors to ensure that the assay reflects true activity of the specific respiratory chain enzymology complex being assayed, i.e., rotenone-sensitive CI activity. In order to avoid misinterpretation, the mitochondrial marker enzyme citrate synthase is typically assayed to enable correction for mitochondrial content by expressing enzymes as citrate synthase ratios. Complex I activity should be determined using the kinetic spectrophotometric assay outlined in Frazier.

We appreciate the reviewer's comment. To answer the concern raised by the reviewer, we also employed kinetic spectrophotometric assays following the protocol detailed by Frazier AE et al as suggested by the reviewer³ (Frazier AE. et al., Methods Cell Biol 2020;155:121-156). As with our previously shown CI enzymatic activity assays, the results from these kinetic spectrophotometric assays reveal a reduction in rotenone-sensitive CI enzymatic activity within podocyte mitochondria derived from *Ins2^{Akita/+}* as well as in *Lep^{db/db}* compared to control mice (Fig 1d). However, in regards to normalization using citrate synthase activity, it is worth noting that while using citrate synthase may be suitable in other conditions, our mitochondrial proteomic analysis indicate a significant reduction in citrate synthase abundance in podocytes from *Ins2^{Akita/+}* mice compared to those from WT mice (*Ins2^{Akita/+}* to WT ratio: 0.63). Western blotting also showed similar results (shown below). Therefore, relying on citrate synthase to normalize measurements in the diabetic condition could potentially lead to an overestimation. Consequently, we opted for an alternative approach, normalizing our data using total mitochondrial protein input as recommended by the Frazier's protocol. Of note, our initial assessment of CI activity involved NADH oxidase staining on frozen tissue sections obtained from mouse kidneys. This approach, widely utilized in the analysis of CI in frozen tissue samples^{20,21} (Kruse SE. et al., Cell Metab 2008;7(4):312-20 and Salagre D. et al., Antioxidants 2023;12(8):1499), served as an initial means to assess CI activity. For a more precise measurement of CI, we employed

mitochondria from primary podocytes placed on a 96-well assay plate precoated with cocktail antibodies against CI subunits. This method is also a widely accepted approach⁴⁻⁶ (Wang T et al., Cell Metab 2021;33(3)531-546.e9; Balsa E et al., Mol Cell 2019;74(5):877-890.e6; and Cao LL et al., Nature 2016 ;539(7630):575-578). Nevertheless, we agree with the reviewer that using different established methods will add to the validity of our experimental approach.

Comment 3: “We also evaluated NDUFS4 staining in glomeruli from diabetic subjects with a wide spectrum of DKD histology, and found that NDUFS4 staining in glomeruli was progressively reduced with worsening of DKD histology (test for trend $P < 0.01$) (Fig. 1I).” Was there a significant difference from Class I CKD to Class III CKD, or between any of the DKD classes? Can you explain the Test for trend? Is it really a progressive decrease in glomerular Ndufs4 staining?

When specifically comparing different Classes of DKD, there is a significant difference in glomerular NDUFS4 staining between Class I and Class IV kidneys, where the difference reached statistical significance ($P = 0.007$) (see figure below). To explore the trend further, we conducted a linear trend analysis. A linear trend analysis, also known as a test for linear trend^{22,23} (Nowak N. et al., Kidney Int 2016; 89:459-467 and Sakaguchi Y. et al., J Am Soc Nephrol 2018; 29:991-999), is a statistical method used to examine whether there is a “systematic and linear relationship between a set of ordered groups or categories and a measured variable.” It is often applied when there is a natural order or progression in the groups being analyzed. The key idea behind linear trend analysis is to assess whether there is a consistent change in the variable of interest as you move from one group to the next. This change is evaluated to determine if it follows a linear pattern, meaning that as you go from one group to the next (e.g., from low to high levels of a variable or from one category to another), there is a consistent increase or decrease in the variable. Based on the results of the trend analysis, we conclude that glomerular NDUFS4 staining exhibits a progressive decline corresponding to the increase in glomerular pathological class of DKD, providing valuable insights into the disease progression.

Comment 4: Fig1k. Statistical analysis was performed on a sample size of $n = 2$ within the microalbuminuria group. How is this possible when using one way-analysis of variance followed by Tukey’s multiple comparisons test in Graphpad Prism? Please confirm the statistical test used.

We acknowledge the limited statistical power within the microalbuminuria group, although it's worth noting that GraphPad did provide the capability to perform a one-way ANOVA followed by Tukey's-Kramer multiple comparison test (<https://www.graphpad.com/support/faqid/591/>). However, following consultation with our biostatistician and in response to the comment by the reviewer, we have made the decision to combine all DKD patients with albuminuria into a single group. Consequently, we have updated Figure 1k to reflect this modification.

Comment 5: What was the mitochondrial density of the podocytes in diabetic mice? *Ins2Akita* and *dbdb* mice.

Using mitochondrial DNA copy number as an indicator of mitochondrial density, we have generated additional data that demonstrate reduced mitochondrial density in podocytes from *Ins2^{Akita/+}* mice when compared to their wild-type (WT) counterparts. We have previously established a similar pattern, illustrating lower mitochondrial copy numbers in podocytes from *Lepr^{db/db}* mice in comparison to their *Lepr^{db/+}* littermates¹⁶ (Long J. et al., J Clin Invest 2016; 126(11):4205)

Comment 6: Fig 3n, ATP production. What are the units of the ATP assay? Both the mitosox and ATP data are expressed as relative to control. As is the Complex I. The ATP assay used appears to be a one-step single reagent assay for detection of viable cells in culture, rather than an assay which can reliably determine ATP production over time in a sensitive manner. I would caution against using such an assay when screening for differences between groups. Was a standard curve generated and used? Did the authors account for cell number or total protein? The same questions apply to the mitosox assay and data. Representative histograms showing mean intensity of MitoSOX fluorescence should be shown for each group.

In the ATP assay, we made a standard curve to measure ATP concentration. In addition, we generated a standard curve to obtain cell numbers based on the DNA concentration using CyQUANT Cell proliferation Assay Kit (Molecular Probes). In response to the reviewer's comments, however, we also performed a real-time label-free ATP assay using Seahorse Analyzer in live primary podocytes to assess real-time total cellular and mitochondrial ATP production rates. As shown in Fig. 3p,q, total and mitochondrial ATP production rates were significantly reduced in podocytes from *Ins2^{Akita/+}* mice, while were normalized in podocytes from *Ins2^{Akita/+}-Ndufs4^{podTg}* mice. Regarding ATP content, MitoSOX and CI enzymatic activity, please refer to our response to comments 3, 10, and 11 to Reviewer #1.

Comment 7: Some of the individual data points on graphs are missing through the manuscript (e.g., Fig 2h, k, l & m; Figure 3g, i, j, k, l) and throughout the paper.

We did not originally include the data points in some of the figures since our understanding was that data points should only be shown when $n < 10$ based on the Journal's guidelines. However, we have added the data points in those figures as well in response to the reviewer.

Comment 8: Fig 2a. What sample is contained in each lane in the genotyping gel?

Sample labels have been provided below the gel image for your reference. To clarify further, the first three lanes represent PCR results from three wild type (WT) samples, while the last three lanes correspond to PCR results from three *Ndufs4^{podTg}* mice.

Comment 9: The mRNA expression of *Ndufs4* appears higher in the "podocyte depleted" fraction of renal cortex vs the podocyte fraction. It is important to show *Ndufs4* protein in podocytes vs tubules and total renal cortex.

The expression levels of *Ndufs4* mRNA are indeed lower in podocytes compared to the podocytes-depleted fraction, which predominately consists of kidney tubules. In the revised version of this manuscript, we have included Western blot data illustrating the levels of *NDUFS4* in both podocytes and tubules (Extended Data Fig. 1h). Additionally, we have mentioned in the text (page 5) that the *NDUFS4* protein expression in kidney tubular cells

remains unchanged in diabetic mice. It is important to highlight that the mitochondrial density in tubules greatly exceeds that in podocytes. This difference in mitochondrial density may account for the elevated expression of Ndufs4 in tubules in comparison to podocytes.

Comment 10: For the Seahorse cell culture studies, how many independent cell culture experiments are shown? It is stated that there were n=6-8 replicates per group. Are these data derived from one cell culture experiment?

The original results were derived from a single cell culture experiment, in which primary podocytes were isolated and pooled from three mice in each experimental group. We have performed similar experiments with primary podocytes isolated from an additional three mice in each group. The results from these additional experiments corroborate with our initial findings, reinforcing the consistency and reliability of our results.

Comment 11: Extended data figure 5a. The western blots are not convincing. The quantitation needs to be shown. In mouse kidney, OPA1 generally has 5 isoforms. There should be 5 bands present. Samples need to be run on a gradient gel to reveal the five isoforms as per PMID: 26822084.

We have incorporated the quantification results obtained from Western blots and conducted an analysis of OPA1 using gradient gel electrophoresis. This analysis was performed on primary podocytes isolated from four distinct groups of mice (Extended Data Figure 5a). Of note, OPA1 does not appear to exhibit 5 distinct isoform bands, possibly due to cell or tissue-specific variations in isoform patterns.

Comment 12: Page 4, line 13. "ETC dysfunction is recognized as an important cause of organ failure in several human pathologies including heart failure, diabetes, and neurodegeneration in a tissue-specific manner". Mitochondrial cytopathies have long been recognized to lead to kidney failure. This should be mentioned in the introduction (PMID: 33305107).

We thank the reviewer and have added these important citations to the manuscript^{24,25} (Schijvens AM et al., *Kidney Int Rep* 2020; 5(12):2146-2159; and Emma F et al., *Nat Rev Nephrol* 2016; 12(5):267-280).

Reviewer #3:

In this manuscript, Mise et al. examine the importance of the mitochondrial ETC Complex I

protein Ndufs4, in the maintenance of mitochondrial structure in the context of the diabetic kidney. The Authors show that Ndufs4 plays a role in the regulation of cristae structure and supercomplex evidence of binding partners in the process. The studies address an interesting topic, and the experimentation appears thorough and well-executed. Specific comments are indicated below.

Comment 1: It is not entirely clear why the Authors chose to focus on Ndufs4. The heat map in Fig.1 shows many ETC proteins, including in Complex I that were also decreased as a result of diabetes. In fact, there are some that were decreased by a greater extent. The decision to choose Ndufs4 comes across as a biased decision without scientific justification. A greater case should be made for choosing this target. In the Discussion, the Authors indicate that examination of other subunits would be meaningful, and this is appreciated, but does not preclude the need for a greater justification for the Ndufs4 focus.

We appreciate the reviewer's comment. Please also refer to our response to Reviewer #1 (comment #4). Briefly, we chose NDUFS4 among several CI subunits in our proteomic experiment for several reasons: Subsequent validation studies in other experimental models of diabetes consistently revealed NDUFS4 as the most prominently downregulated subunit in diabetes when compared to nondiabetic controls, further substantiating its significance. In contrast, several other potential candidates, including NDUFA2, NDUFB3, NDUFB4, NDUFB5, NDUFB8, NDUFB11, and NDUFV3 were not consistently downregulated (Extended Data Fig. 1e,g). Importantly, examination of human samples of individuals with DKD indicated that the downregulation of NDUFS4 occurs prior to the onset of albuminuria in patients, suggesting its potential role as an early marker of dysfunction.

Comment 2: Introduction, page 4; In the last sentence, the Authors state that their "results unexpectedly reveal that ETC integrity determines the stability of RSCs." This seems disingenuous. Do the Authors really believe that the very integrity of the components of the ETC would not be an important factor in the stability of the supercomplex in which they reside? My suggestion would be to either remove or soften this statement.

We appreciate the reviewer's suggestion. In the revised manuscript, we have removed the term "unexpectedly" from the content. Our intention simply was to convey that we did not anticipate that Ndufs4 overexpression would impact not only CI activity, but also RSC stability.

Comment 3: The use of multiple experimental models of diabetes (Akita, db/db) and human patient samples is appreciated and strengthen the conclusions and relevance of the studies. With that being said, the inclusion of complete db/db data (especially proteomic profiling in Fig.1b) is glaringly absent. Inclusion of this data makes a stronger case for commonality in the models and the ultimate focus on Ndufs4.

We thank the reviewer's suggestion and have performed mitochondrial proteomic analysis using mitochondria isolated from podocytes of *Lepr^{db/+}* and *Lepr^{db/db}* mice. The data showed that NDUFS4 was one of the most downregulated among the CI subunits. This gave us stronger support in selecting NDUFS4 in this study. We have included the new data in Fig. 1e and Extended Data Fig. 1d.

REFERENCES

- 1 Cabrera-Orefice, A., Potter, A., Evers, F., Hevler, J. F. & Guerrero-Castillo, S. Complexome profiling-exploring mitochondrial protein complexes in health and disease. *Front. Cell Dev. Biol.* **9**, 796128 (2021).
 - 2 Välikangas, T., Suomi, T. & Elo, L. L. A systematic evaluation of normalization methods in quantitative label-free proteomics. *Brief Bioinform.* **19**, 1-11 (2018).
 - 3 Frazier, A. E., Vincent, A. E., Turnbull, D. M., Thorburn, D. R. & Taylor, R. W. Assessment of mitochondrial respiratory chain enzymes in cells and tissues. *Methods Cell Biol.* **155**, 121-156 (2020).
 - 4 Wang, T. *et al.* C9orf72 regulates energy homeostasis by stabilizing mitochondrial complex I assembly. *Cell Metab.* **33**, 531-546.e539 (2021).
 - 5 Balsa, E. *et al.* ER and nutrient stress promote assembly of respiratory chain supercomplexes through the PERK-eIF2alpha axis. *Mol. Cell* **74**, 877-890.e876 (2019).
 - 6 Cao, L. L. *et al.* Control of mitochondrial function and cell growth by the atypical cadherin Fat1. *Nature* **539**, 575-578 (2016).
 - 7 Falkevall, A. *et al.* Reducing VEGF-B Signaling Ameliorates Renal Lipotoxicity and Protects against Diabetic Kidney Disease. *Cell Metab.* **25**, 713-726 (2017).
 - 8 Wang, W. *et al.* Mitochondrial fission triggered by hyperglycemia is mediated by ROCK1 activation in podocytes and endothelial cells. *Cell Metab.* **15**, 186-200 (2012).
 - 9 Galvan, D. L. *et al.* Drp1S600 phosphorylation regulates mitochondrial fission and progression of nephropathy in diabetic mice. *J. Clin. Invest.* **129**, 2807-2823 (2019).
 - 10 Sutandy, F. X. R., Gößner, I., Tascher, G. & Münch, C. A cytosolic surveillance mechanism activates the mitochondrial UPR. *Nature* **618**, 849-854 (2023).
 - 11 Labuschagne, C. F., Cheung, E. C., Blagih, J., Domart, M. C. & Vousden, K. H. Cell Clustering Promotes a Metabolic Switch that Supports Metastatic Colonization. *Cell Metab.* **30**, 720-734.e725 (2019).
 - 12 Murphy, M. P. *et al.* Guidelines for measuring reactive oxygen species and oxidative damage in cells and in vivo. *Nat Metab* **4**, 651-662 (2022).
 - 13 Galvan, D. L. *et al.* Real-time in vivo mitochondrial redox assessment confirms enhanced mitochondrial reactive oxygen species in diabetic nephropathy. *Kidney Int.* **92**, 1282-1287 (2017).
 - 14 Waypa, G. B. *et al.* Hypoxia triggers subcellular compartmental redox signaling in vascular smooth muscle cells. *Circ. Res.* **106**, 526-535 (2010).
 - 15 Moullan, N. *et al.* Tetracyclines Disturb Mitochondrial Function across Eukaryotic Models: A Call for Caution in Biomedical Research. *Cell Rep.* **10**, 1681-1691 (2015).
 - 16 Long, J. *et al.* Long noncoding RNA Tug1 regulates mitochondrial bioenergetics in diabetic nephropathy. *J. Clin. Invest.* **126**, 4205-4218 (2016).
 - 17 Qi, W. *et al.* Pyruvate kinase M2 activation may protect against the progression of diabetic glomerular pathology and mitochondrial dysfunction. *Nat. Med.* **23**, 753-762 (2017).
 - 18 Fu, Y. *et al.* Elevation of JAML Promotes Diabetic Kidney Disease by Modulating Podocyte Lipid Metabolism. *Cell metabolism* **32**, 1052-1062.e1058 (2020).
 - 19 Cao, A. *et al.* DACH1 protects podocytes from experimental diabetic injury and modulates PTIP-H3K4Me3 activity. *The Journal of clinical investigation* **131** (2021).
 - 20 Kruse, S. E. *et al.* Mice with mitochondrial complex I deficiency develop a fatal encephalomyopathy. *Cell Metab.* **7**, 312-320 (2008).
 - 21 Salagre, D., Raya Álvarez, E., Cendan, C. M., Aouichat, S. & Agil, A. Melatonin Improves Skeletal Muscle Structure and Oxidative Phenotype by Regulating Mitochondrial Dynamics and Autophagy in Zucker Diabetic Fatty Rat. *Antioxidants (Basel)* **12** (2023).
 - 22 Nowak, N. *et al.* Increased plasma kidney injury molecule-1 suggests early progressive renal decline in non-proteinuric patients with type 1 diabetes. *Kidney Int* **89**, 459-467 (2016).
-

- 23 Sakaguchi, Y., Hamano, T., Wada, A., Hoshino, J. & Masakane, I. Magnesium and Risk of Hip Fracture among Patients Undergoing Hemodialysis. *Journal of the American Society of Nephrology : JASN* **29**, 991-999 (2018).
 - 24 Schijvens, A. M. *et al.* Mitochondrial Disease and the Kidney With a Special Focus on CoQ(10) Deficiency. *Kidney Int. Rep.* **5**, 2146-2159 (2020).
 - 25 Emma, F., Montini, G., Parikh, S. M. & Salviati, L. Mitochondrial dysfunction in inherited renal disease and acute kidney injury. *Nat. Rev. Nephrol.* **12**, 267-280 (2016).
-

REVIEWER COMMENTS

Reviewer #1 (Remarks to the Author):

I highly appreciate the inclusion of additional experiments, as well as the thorough rebuttal, who have taken away several of my concerns. However, I feel that some of my previous suggestions deserve further attention:

1. Regarding Fig. 1b: I appreciate the new table but still cannot easily determine if these changes are significant; please include p-values in the new table (Extended data Fig. 1d)

2. Although the authors present some arguments (e.g. their reply to comment 4, 6 and 17), I'm still not convinced that the observed effects are Ndufs4-specific. The accessory NDUFS4 protein is one of the essential CI subunits. Unless it has an additional function (perhaps the role described in this manuscript?), a drop in NDUFS4 protein levels will always induce a drop in the total protein level of assembled CI. The latter will induce a drop in protein level of virtually all CI subunits. In my opinion, the described effects all can be explained by a reduction in the level of fully assembled CI and therefore cannot be regarded as ndufs4-specific?

3. Extended data Fig. 3b: the y-axis title contains an error. Moreover, in the main text the authors state that: "...ECAR... also showed similar changes..". What does this mean? If there really is a drop in OCR, I would expect an increase in ECAR? As described in the literature, ECAR is not exclusively a measure of glycolytic rate (e.g. lactate production); please discuss the OCR and ECAR results in an integrated manner.

4. Regarding my previous comments 8, 9, 12 and 14: please ensure that the info provided in the author's rebuttal is included in the manuscript.

5. I appreciate the use of roGFP (I guess it is roGFP1?). This contains an S=S bridge, which is broken or formed based upon its thiol redox environment. Please interpret the results obtained with this sensor as a measure of thiol redox state (this of course is hydrogen peroxide sensitive in most systems).

6. Regarding my previous comment 15. The Extended data Fig. 4h referred to in the rebuttal is not on supercomplexes but on ATP levels?

7. Regarding the NDUFS4-STOML2 interaction. In the rebuttal there is referred to Extended data Fig. 4g, but this panel has nothing to do with STOML2. I do not understand the choice for 5XTB, which represents only part of the CI matrix arm. Please investigate whether the STOML2 binding sites on the NDUFS4 protein are accessible when NDUFS4 is incorporated in fully assembled CI and the latter is part of the RSC. This will provide insight on whether STOML2 can bind to NDUFS4 when the latter is within the ETC supercomplex.

Reviewer #2 (Remarks to the Author):

The authors have responded to all of my queries satisfactorily. The manuscript is much improved.

Reviewer #3 (Remarks to the Author):

The Authors have addressed my concerns.

Reviewer #1

Comment 1: Regarding Fig. 1b: I appreciate the new table but still cannot easily determine if these changes are significant; please include p-values in the new table (Extended data Fig. 1d)

We isolated podocytes from 8 wild type (WT) and 8 diabetic *Ins2^{Akita/+}* mice (Fig1. Legend, a). Subsequently, we pooled all the podocytes within each group for a comprehensive screening of differentially expressed mitochondrial subunits in our proteomic analysis. While this pooling strategy might have limited the statistical significance of individual subunits when comparing WT and diabetic mice, we addressed this limitation by incorporating and validating our results using additional approaches, including qRT-PCR analyses (Extended Data Fig. 1e-g) and using an alternative diabetic *Lepr^{db/db}* mouse model.

Comment 2: Although the authors present some arguments (e.g., their reply to comment 4, 6 and 17), I'm still not convinced that the observed effects are Ndufs4-specific. The accessory NDUF54 protein is one of the essential CI subunits. Unless it has an additional function (perhaps the role described in this manuscript?), a drop in NDUF54 protein levels will always induce a drop in the total protein level of assembled CI. The latter will induce a drop in protein level of virtually all CI subunits. In my opinion, the described effects all can be explained by a reduction in the level of fully assembled CI and therefore cannot be regarded as Ndufs4-specific?

Our findings strongly suggest that the observed effects on mitochondrial and kidney function described in this study are indeed Ndufs4-specific since we specifically targeted Ndufs4 in our experimental approach. However, it is important to clarify that while our results support the critical role of Ndufs4 in the observed effects, this does not imply exclusivity. Our findings suggest that Ndufs4 overexpression through its interaction with STOML2 stabilizes cristae morphology, providing a platform for assembly of the mitochondrial respiratory chain complexes. However, we recognize that these findings do not negate the potential role of other CI subunits in sustaining CI integrity and enhancing overall mitochondrial function. Indeed, we concur with the reviewer that improving CI and mitochondrial dysfunction in the diabetic environment may not be confined solely to Ndufs4, recognizing the plausible involvement of other CI subunits in CI integrity and improved mitochondrial function. We have highlighted this point in the Discussion.

Comment 3: Extended data Fig. 3b: the y-axis title contains an error. Moreover, in the main text the authors state that: "...ECAR... also showed similar changes.". What does this mean? If there really is a drop in OCR, I would expect an increase in ECAR. As described in the literature, ECAR is not exclusively a measure of glycolytic rate (e.g., lactate production); please discuss the OCR and ECAR results in an integrated manner.

We appreciate the reviewer's comment. It seems that an error occurred during the PDF conversion, and original 10^5 was mistakenly displayed as 10^{\square} . We have now rectified this error in the revised version. Regarding the interpretation of ECAR results, the interplay between OCR and ECAR in diabetic kidney disease is indeed complex. Consistent with our findings, a recent publication has provided evidence that chronic hyperglycemia results in lower ECAR and OCR levels in podocytes¹. However, we took the reviewer's comment as an

opportunity to further clarify the relationship between aberrant OCR and ECAR in our experimental model. We have now added a new figure (Extended Data Fig. 3c), elucidating the energy maps of primary podocytes isolated from the four distinct experimental groups of mice by combining OCR and ECAR data. The overall goal was to understand how the podocytes energy metabolism adapts and changes under diabetic environment. This graphical representation suggests that podocytes derived from diabetic *Ins2^{Akita/+}* mice are inefficient in energy production in diabetic conditions, whereas overexpression of *Ndufs4* in diabetic *Ins2^{Akita/+};Ndufs4^{PodTg}* mice exhibit an energy map similar to the podocytes from WT mice, indicating that they use both mitochondrial respiration and glycolysis for energy production.

Comment 4: Regarding my previous comments 8, 9,12 and 14: please ensure that the info provided in the author's rebuttal is included in the manuscript.

We have incorporated additional information based on our response to the reviewer's comments in the revised manuscript. Most comments were already present in the previous revision manuscript and thereby are no longer highlighted in the current revised version. Please refer to pages 4, 9, 25, and 38.

Comment 5: I appreciate the use of roGFP (I guess it is roGFP1?). This contains an S=S bridge, which is broken or formed based upon its thiol redox environment. Please interpret the results obtained with this sensor as a measure of thiol redox state (this of course is hydrogen peroxide sensitive in most systems).

As previously described by our group and others^{2,3}, we used mito-roGFP2, a mitochondrial matrix-targeted ratiometric redox-sensitive green fluorescent protein. This sensor contains two adjacent cysteine residues that do not interact in reduced thiol redox state where the sensor exhibits high emission at 525 nm when excited at 488 nm, and relatively low emission at 525 nm when excited at 405 nm. Protein thiol oxidation mediated by H₂O₂ leads to the formation of a disulfide linkage, resulting in a conformational change in the sensor that increases emission at 525 nm during excitation at 405 nm, and decreases emission at 525 nm when excited at 488 nm^{2,3}. We measured mito-roGFP ratios in live mitochondria using confocal microscopy, in live cells. We found that primary podocytes from diabetic mice showed an increase in oxidized/reduced mito-roGFP ratios compared to those from WT mice (Fig. 3o), suggesting that mitochondrial thiol oxidation increases in diabetic podocytes. On the other hand, primary podocytes from diabetic mice overexpressing *Ndufs4* show significantly reduced oxidized/reduced mito-roGFP ratios compared with podocytes from diabetic mice, indicating *Ndufs4* overexpression prevents enhanced mROS in diabetes. We added this information in the text and Methods.

Comment 6: Regarding my previous comment 15. The Extended data Fig. 4h referred to in the rebuttal is not on supercomplexes but on ATP levels?

In the revised manuscript, the data are shown in Extended data Fig. 4i.

Comment 7: Regarding the NDFUS4-STOML2 interaction. In the rebuttal there is referred to Extended data Fig. 4g, but this panel has nothing to do with STOML2. I do not understand the choice for 5XTB, which represents only part of the CI matrix arm. Please investigate whether the STOML2 binding sites on the NDUFS4 protein are accessible when NDUFS4 is

incorporated in fully assembled CI and the latter is part of the RSC. This will provide insight on whether STOML2 can bind to NDUFS4 when the latter is within the ETC supercomplex.

In the revised version of the manuscript, the Extended Data Fig. 4j provides evidence that the CI is present almost exclusively in the form of stable RSCs in podocytes.

The structural information for macromolecules containing Ndufs4 was retrieved from the Protein Data Bank (PDB; <https://www.rcsb.org/>). There are four entries in PDB with macromolecules containing NDUFS4: 5XTB (matrix arm of CI), 5XTC (CI), 5XTH (RSC with I₁III₂IV₁ stoichiometry), and 5XTI (I₂III₂IV₂ stoichiometry), each with a slightly different resolution. We chose PDB-5XTB for further study because of its highest resolution (3.40Å). To address whether NDUFS4 protein is accessible to STOML2 in the context of RSC, we used the molecular structure visualization and analysis tool ChimeraX (UCSF ChimeraX, <https://www.cgl.ucsf.edu/chimerax/>), a powerful molecular modeling engine, to analyze the structure of the I₁III₂IV₁ within RSC resolved by CryoEM (PDB-5XTH). This program shows that CIII and CIV bind to the hydrophobic membrane arm of CI, forming the RSC. Furthermore, it reveals that the NDUFS4 protein (magenta, Extended Data Fig. 6c) is localized on the surface of the matrix arm of the CI within the RSC, exposing NDUFS4 to potential interactions with other molecules such as STOML2. Our experimental data with STOML2 knockout cells and STOML2 mutation studies provide additional support to this interaction. Overall, these findings shed light on potential interactions involving NDUFS4 and STOML2. The figure was added to the manuscript (Extended Data Fig. 6c).

References:

1. Qi, W. *et al.* Pyruvate kinase M2 activation may protect against the progression of diabetic glomerular pathology and mitochondrial dysfunction. *Nat. Med.* 23, 753-762 (2017).
2. Waypa, G. B. *et al.* Hypoxia triggers subcellular compartmental redox signaling in vascular smooth muscle cells. *Circ. Res.* 106, 526-535 (2010).
3. Galvan, D. L. *et al.* Real-time in vivo mitochondrial redox assessment confirms enhanced mitochondrial reactive oxygen species in diabetic nephropathy. *Kidney Int.* 92, 1282-1287 (2017).

REVIEWERS' COMMENTS

Reviewer #1 (Remarks to the Author):

I thank the authors for their rebuttal, which have addressed my remaining concerns.